# FALCON: FEW-STEP ACCURATE LIKELIHOODS FOR CONTINUOUS FLOWS

**Danyal Rehman**[1,2,3]\***Tara Akhound-Sadegh**[1,4]**, Artem Gazizov**[5]**, Yoshua Bengio**[1,2,6]**,
Alexander Tong**[3]
[1]Mila – Québec AI Institute, [2]Université de Montréal, [3]AITHYRA,
[4]McGill University, [5]Harvard University, [6]CIFAR Senior Fellow

## ABSTRACT

Scalable sampling of molecular states in thermodynamic equilibrium is a long-standing challenge in statistical physics. Boltzmann Generators tackle this problem by pairing a generative model, capable of exact likelihood computation, with importance sampling to obtain consistent samples under the target distribution. Current Boltzmann Generators primarily use continuous normalizing flows (CNFs) trained with flow matching for efficient training of powerful models. However, likelihood calculation for these models is extremely costly, requiring thousands of function evaluations per sample, severely limiting their adoption. In this work, we propose FEW-STEP ACCURATE LIKELIHOODS FOR CONTINUOUS FLOWS (FALCON), a method which allows for few-step sampling with a likelihood accurate enough for importance sampling applications by introducing a hybrid training objective that encourages invertibility. We show FALCON outperforms state-of-the-art normalizing flow models for molecular Boltzmann sampling and is over *two orders of magnitude faster* than the equivalently performing CNF model. FALCON code is available at https://github.com/danyalrehman/FALCON.

## 1 INTRODUCTION

Sampling molecular configurations from the Boltzmann distribution $p(x) \propto \exp(-\mathcal{E}(x))$ where $\mathcal{E}(x)$ is the potential energy of a configuration $x$, is a foundational and long-standing challenge in statistical physics. The ability to generate samples according to this distribution is the foundation for determining many other observables—such as free energies and heat capacities—which govern real-world behaviour. Consequently, efficient Boltzmann sampling is essential for progress in a large range of areas, from characterizing the function of biomolecules, to accelerating drug design, and discovering novel materials (Frenkel & Smit, 2023; Liu, 2001; Ohno et al., 2018; Stoltz et al., 2010).

The difficulty of this task arises from the structure of the energy for molecules of interest. The energy landscape is high-dimensional and non-smooth with many local energy minima. These rugged energies severely challenge classical simulation-based methods like Molecular Dynamics (MD) (Leimkuhler & Matthews, 2015) and Monte Carlo Markov Chains (MCMC) as they become easily trapped in local minima, requiring a computationally inaccessible number of steps to mix between modes. These samplers generate many correlated samples, creating large inefficiencies, where an ideal sampler would generate i.i.d. samples from the underlying data distribution, $p(x)$.

Boltzmann Generators (BGs) have emerged as a way to address this inefficiency by amortizing the cost through the training of a generative model to learn to sample from $p_\theta(x)$ close to $p(x)$. These samples can then be corrected to $p(x)$ using self-normalized importance sampling (SNIS) (Noé et al., 2019). SNIS requires efficient access to $p_\theta(x)$ and $\mathcal{E}(x)$ for every sample drawn $x \sim p_\theta(x)$ for the correction step, but guarantees statistical consistency of the corrected samples, as illustrated in Fig. 1 (Noé et al., 2019; Liu, 2001).

The main design choice in BGs is which type of generator to use. Modern BGs (Klein et al., 2023; Klein & Noe, 2024) primarily make use of generators based on continuous normalizing flows (CNFs) (Chen et al., 2018; Grathwohl et al., 2019) due to their expressive power, ease of training,

---

\*Correspondence to danyal.rehman@mila.quebec, atong@aithyra.at

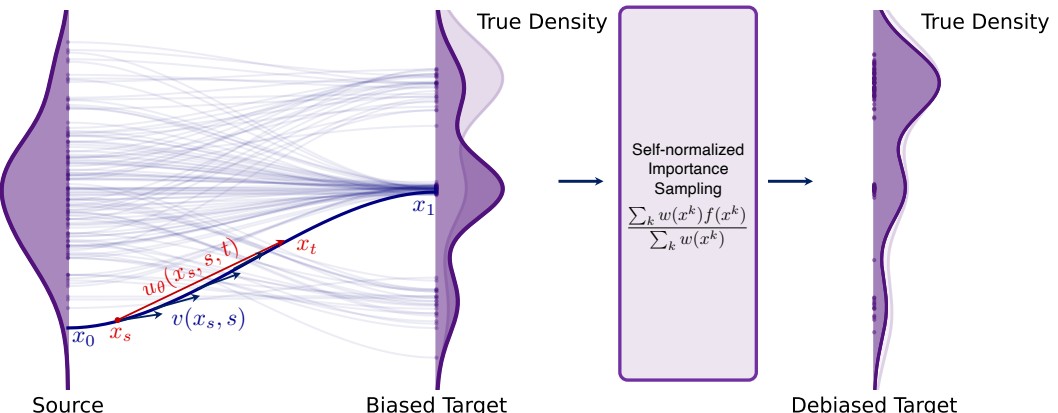

Figure 1: Flow map learns from biased data, with SNIS re-weighting generated samples consistent with the Boltzmann distribution, approaching equality with infinite samples under mild regularity conditions.

and flexibility of parameterization (Köhler et al., 2020) (see Table 1). However, while it is possible to access the $p_\theta(x)$ of a CNF, it is extremely computationally costly to approximate $p_\theta(x)$ with sufficient accuracy. Two primary reasons contribute to this cost: (**1**) Approximate estimators are not sufficiently accurate, making full Jacobian calculations necessary for each step along the flow; and (**2**) Many steps are necessary to control discretization error for sufficient performance (Fig. 2).

Recently, there have been significant advancements in few-step generation using flow models (Song et al., 2023; Boffi et al., 2025a; Frans et al., 2025; Guo et al., 2025; Sabour et al., 2025; Geng et al., 2025a). These models are extremely powerful few-step generators with flexible architectures and efficient simulation-free training; however, these few-step samplers do not natively admit efficient estimators of the likelihood, making them unsuitable for the high-precision demands of importance sampling and scientific applications such as Boltzmann Generation (Rehman et al., 2025).

In this work, we investigate how to design a generative model that combines the best of both worlds: the training efficiency and architectural freedom of simulation-free flow models with the fast sampling and likelihood evaluation of discrete-time invertible models. We propose

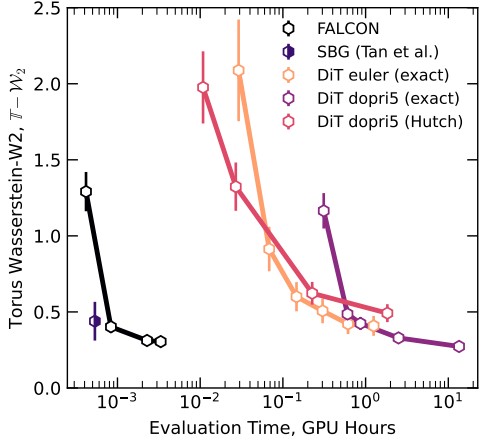

Figure 2: Performance-inference time comparison between NFs and CNFs for $10^4$ dipeptide samples.

FEW-STEP ACCURATE LIKELIHOODS FOR CONTINUOUS FLOWS (FALCON), a flow-based model that enables few-step sampling while providing a likelihood estimate that is both fast to compute and accurate enough for importance sampling applications. FALCON leverages a hybrid training objective that combines a regression loss for stable and efficient few-step generation with a cycle-consistency term to encourage invertibility prior to convergence. Our main contributions are:

- We introduce FEW-STEP ACCURATE LIKELIHOODS FOR CONTINUOUS FLOWS (FALCON): a new continuous flow-based generative model for Boltzmann sampling that is invertible, trainable with a regression loss, and supports free-form architectures, while enabling both few-step generation and efficient likelihood evaluation.

- Orthogonally, we introduce a simple and scalable, softly equivariant continuous flow architecture that significantly improves over the current state-of-the-art equivariant flow model architecture.

- We show that FALCON is **two orders of magnitude faster** than CNF-based Boltzmann Generators for equivalent performance (Fig. 2), drastically reducing the computational cost of CNFs, and taking significant strides towards real-world large-scale molecular sampling applications.

- We show that FALCON outperforms the current state-of-the-art normalizing flow-based Boltzmann Generator across all metrics, even when FALCON is given **250× fewer samples** (Tan et al., 2025a).

## 2 BACKGROUND AND PRELIMINARIES

We are interested in drawing statistically independent samples from a target Boltzmann distribution $p_{\text{target}}$ with partition function $\mathcal{Z}$, defined over $\mathbb{R}^d$:

$$p_{\text{target}}(x) \propto \exp\left(-\mathcal{E}(x)\right), \quad \mathcal{Z} = \int_{\mathbb{R}^d} \exp\left(-\mathcal{E}(x)\right) dx \tag{1}$$

where $\mathcal{E} : \mathbb{R}^d \to \mathbb{R}$ is the energy of the system, which we can efficiently compute for any $x$. In this work we do not require the energy to be differentiable. Unlike in the pure sampling setting (Akhound-Sadegh et al., 2024; Havens et al., 2025; Akhound-Sadegh et al., 2025; Midgley et al., 2023; Zhang & Chen, 2022; Vargas et al., 2023), we also assume access to a small biased dataset $\mathcal{D} = \{x^i\}_{i=1}^N$ of $N$ samples (Noé et al., 2019). This makes it possible to perform an initial learning phase that fits a generative model with parameters $\theta$, producing a proposal distribution, $p^\theta(x)$ (Noé et al., 2019).

**Boltzmann Generators.** (BGs) (Noé et al., 2019) combine generative models capable of exact likelihoods, with a target energy function and self-normalized importance sampling (SNIS) to re-weight generated samples to the target Boltzmann distribution. The model is first trained on a possibly biased dataset $\mathcal{D}$ as close as possible to $p_{\text{target}}$. BGs then draw $K$ independent samples $x^i \sim p_1^\theta, i \in [K]$ and compute the corresponding unnormalized importance weights such that $w(x^i) \triangleq \exp(-\mathcal{E}(x^i))/p_1^\theta(x^i)$. Given these importance weights, we can then compute a consistent Monte–Carlo estimate of any observable $o(x)$ under $p_{\text{target}}$ using SNIS (Liu, 2001) as:

$$\mathbb{E}_{p_{\text{target}}}[o(x)] = \mathbb{E}_{p_1^\theta}[o(x)\bar{w}(x)] \approx \frac{\sum_{i=1}^K w(x^i)o(x^i)}{\sum_{i=1}^K w(x^i)}. \tag{2}$$

This allows for inference-time scaling as the Monte–Carlo estimate of any observable converges in probability to the correct value as the number of samples grows.

**Flow Matching Models.** Flow matching models (Lipman et al., 2023; Albergo & Vanden-Eijnden, 2023; Liu, 2022; Peluchetti, 2021) are probabilistic generative models that learn a continuous interpolation between an easy-to-sample distribution $p_0 = p_{\text{noise}}$ and the data distribution $p_1 = p_{\text{data}}$ in $\mathbb{R}^d$. Let $x_s = sx_1 + (1-s)x_0$ be a point at time $s \in [0, 1]$ between two points $x_0 \sim p_0$ and $x_1 \sim p_1$. The flow matching objective is then $\mathbb{E}_{x_0 \sim p_0, x_1 \sim p_1, s \sim \text{Unif}(0,1)} w(s) \|v_\theta(x_s, s) - (x_1 - x_0)\|_2^2$ for some parameterized vector field $v_\theta$, and weighting function $w : [0, 1] \to \mathbb{R}^+$. We can then sample using an ordinary differential equation (ODE) $x_s^\theta = \int_0^s v_\theta(x_\tau, \tau)d\tau$ with the initial condition $x_0 \sim p_0$ and are guaranteed (under some mild assumptions) that $p_1^\theta(\hat{x}_1) \approx p_1(\hat{x}_1)$.

Furthermore, the density $p_s^\theta$ can be computed using the instantaneous change of variables formula (Chen et al., 2018) $\frac{\partial \log p(x_s)}{\partial s} = -\text{tr}\left(\frac{\partial v_\theta}{\partial x_s}\right)$ by solving a single $d + 1$ dimensional ODE:

$$\begin{bmatrix} x_s \\ \log p_s^\theta(x_s) \end{bmatrix} = \int_0^s \begin{bmatrix} v_\theta(x_\tau, \tau) \\ -\text{tr}\left(\frac{\partial v_\theta}{\partial x_\tau}\right) \end{bmatrix} d\tau, \text{ with initial condition } \begin{bmatrix} x_0 \\ \log p_0(x_0) \end{bmatrix} \tag{3}$$

where the integral is discretized into $T$ steps and the trace can either be computed exactly in $O(dT)$ function evaluations or approximated using Hutchinson's trace estimator $\text{tr}(J) = \mathbb{E}_\epsilon[\epsilon^T J \epsilon]$ for some noise vector $\epsilon \in \mathbb{R}^d$ in $O(T)$ function evaluations (Hutchinson, 1990). In practice, this is a major bottleneck because a large number of steps is needed to control discretization error (see Fig. 2).

**Few-step Flow Models.** Flow matching models can require hundreds of steps to accurately approximate $p_1$. To speed up sampling, few-step flow models such as consistency models (CMs) (Song et al., 2023), optimal transport-based

Table 1: Related method overview

| Method | Invertible | Regression-loss | Few Step | FreeForm Arch. |
|---|---|---|---|---|
| BioEmu | ✗ | ✓ | ✗ | ✓ |
| FlowMaps | ✗ | ✓ | ✓ | ✓ |
| TBG | ✓ | ✓ | ✗ | ✓ |
| RegFlow | ✓ | ✓ | ✓ | ✗ |
| Prose | ✓ | ✗ | ✓ | ✗ |
| FFFlows | ✓ | ✗ | ✓ | ✓ |
| FALCON (Ours) | ✓ | ✓ | ✓ | ✓ |

methods (Pooladian et al., 2023; Tong et al., 2024a;b; Shi et al., 2023), and flow map models (Boffi et al., 2025a; Sabour et al., 2025; Geng et al., 2025a; Frans et al., 2025; Guo et al., 2025) attempt to

train a model that generates high quality samples in many fewer steps. Recently, efficient models that take not only the current sample time, but also the target sample time have shown flexibility and effectiveness in the one- to few-step regimes. In these models, $u_\theta$ is augmented with an additional input $t$, which denotes the target time to capture the *average velocity* (Geng et al., 2025a) $u$ as:

$$u(x_s, s, t) = \frac{1}{t-s} \int_s^t v(x_\tau, \tau) d\tau \tag{4}$$

to minimize the average velocity objective:

$$\mathbb{E}_{s,t,x_s} \left[ w(s,t) \left\| u_\theta(x_s, s, t) - \frac{1}{t-s} \int_s^t v(x_\tau, \tau) d\tau \right\|^2 \right] \tag{5}$$

where the average velocity $u_\theta$ is parameterized by a neural network as depicted in Fig. 1 and $v$ is the vector field of the probability flow ODE that transports samples from the noise distribution $p_0$ to the data distribution $p_1$. Once the average velocity is learned, then samples can be drawn using any discretization of the time interval $[0, 1]$, such that $t_0 = 0, t_1, \ldots t_T = 1$ as $x_{t_i} = x_{t_{i-1}} + (t_i - t_{i-1})u_\theta(x_{t_{i-1}}, t_{i-1}, t_i)$ for $i \in 1 \ldots T$. However, thus far, few-step flow models have only been applied for fast generation, and, as we show, do not natively guarantee efficient access to likelihoods in realistic settings as the learned average velocity map $u_\theta$ is not guaranteed to be invertible before the training objective is perfectly minimized, making the standard change-of-variables formula inapplicable. These models and their relationship to FALCON are summarized in Table 1.

## 3 FEW-STEP ACCURATE LIKELIHOODS FOR CONTINUOUS FLOWS

We now introduce FEW-STEP ACCURATE LIKELIHOODS FOR CONTINUOUS FLOWS (FALCON), a novel flow-based generative model designed to address the inherent efficiency limitations of using continuous flow models as Boltzmann Generators. Our method departs from traditional continuous normalizing flow (CNFs) by training a flow map that operates in a few discrete steps, while simultaneously achieving invertibility to ensure fast and accurate likelihood computation for Boltzmann Generation. This is achieved through a hybrid training objective, which, by enabling stable few-step generation, dramatically reduces the inference cost. This efficiency allows us to train more expressive architectures (Vaswani et al., 2017; Peebles & Xie, 2023; Ma et al., 2024) that were previously computationally infeasible to scale in the Boltzmann Generator setting.

**Flow Maps are Flawed Boltzmann Generators.** The core of FALCON is a generative process that learns an invertible map from a simple base distribution $p_0$ to the target molecular distribution $p_1$ in a small number of steps. We first examine the suitability of the existing few-step flows for importance sampling applications, concluding that, on their own, they are not sufficient. We first define the continuous map with respect to a vector field $v$ as:

$$X_v(x_s, s, t) = \int_s^t v(x_\tau, \tau) d\tau + x_s, \tag{6}$$

and note that under mild regularity conditions on $v$, this map is always invertible up to discretization error. For any invertible map, we can compute the change in density with respect to the input using the change of variables formula (which requires the Jacobian), at the cost of $d$ function evaluations, and the determinant, which, while an $O(d^3)$ operation, is in practice negligible compared to the NFE cost (See §D.2). This property also holds for flow maps at the optima, which we address in Proposition 1:

**Proposition 1.** *Let $u_\theta^\star$ be a minimizer of Eq. 5 with respect to some $v$. Also, define the Jacobian of $X$ as $\mathbf{J}_X = \frac{\partial X}{\partial x_s}$, and the discrete flow map:*

$$X_u(x_s, s, t) = x_s + (t-s)u_\theta^\star(x_s, s, t) \tag{7}$$

*Then, for sufficiently smooth $u_\theta^\star$ and $v$ and for any $(s, t) \in [0, 1]^2$,*

*1. $X_u(\cdot, s, t)$ is an invertible map everywhere,*

*2. $\log p_t^{u^\star}(x_t) = \log p_s^{u^\star}(x_s) - \log |\det \mathbf{J}_{X_u}(x_s)|$ almost everywhere.*

We provide a more precise statement and proofs for all propositions in §A. This means that *optimal* flow maps are, in some ways, ideal Boltzmann Generators in that they have relatively efficient access to both samples and likelihood; however, this property only holds at the optima, and in the case that $X_u(\cdot, s, t) = X_v(\cdot, s, t)$ for all $s, t$, which in practice is extremely challenging to satisfy.

In practice, for standard flow map models, $X_u(\cdot, s, t) \neq X_v(\cdot, s, t)$ and we have no guarantee that $X_u$ will be invertible, making efficient likelihood calculation all but impossible. However, we note that this condition is actually much stronger than we need for FALCON. For our uses, while we would like $X_u(\cdot, s, t)$ to be close to $X_v(\cdot, s, t)$, for accurate and efficient likelihood computation, we only require that $X_u$ is invertible, not that it matches the particular invertible map defined by $X_v$. This leads us to define an additional invertibility loss:

$$\mathcal{L}_{\text{inv}}(\theta) = \mathbb{E}_{s,t,x_s}\|x_s - X_u(X_u(x_s, s, t), t, s)\|^2, \tag{8}$$

to be used in conjunction with the average velocity objective and flow matching objectives, $\mathcal{L}_{\text{cfm}}$, for a final loss comprised of three components:

$$\mathcal{L}(\theta) = \mathcal{L}_{\text{cfm}}(\theta) + \lambda_{\text{avg}}\mathcal{L}_{\text{avg}}(\theta) + \lambda_r\mathcal{L}_{\text{inv}}(\theta), \tag{9}$$

with variants $\mathcal{L}_{\text{avg}}$ proposed below. Minimizing this loss has a less strict requirement for the correctness of the Boltzmann Generator specifically:

**Proposition 2.** *Let $u_\theta^\star$ be a minimizer of $\mathcal{L}_{inv}$ (Eq. (8)) with respect to some $v$. Then, for sufficiently smooth $u_\theta^\star$ and $v$ and for any $(s, t) \in [0, 1]^2$, $X_u(\cdot, s, t)$ is an invertible map everywhere, and $\log p_t^{u^\star}(x_t) = \log p_s^{u^\star}(x_s) - \log|\det \mathbf{J}_{X_u}(x_s)|$ almost everywhere.*

Thus, minimizing the invertibility loss is sufficient for valid Boltzmann Generation, even without exactly reproducing the continuous-time flow. Note that the proposition provides a constructive guarantee of invertibility, and, in practice, we only require the existence of an inverse, not its explicit form. This condition ensures that FALCON acts as a consistent generator of the target energy distribution, $\mathcal{E}(x)$, while benefiting from fast inference-time scalability.

**FALCON Enables Scalable Architectures.** Previous Boltzmann Generators based on continuous normalizing flows for molecular applications utilize small equivariant architectures (Klein et al., 2023; Klein & Noe, 2024; Tan et al., 2025a; Aggarwal et al., 2025). These models are limited in their scale due to the high cost of inference with multi-step adaptive step size samplers, which are needed to control the error in the likelihood calculation. FALCON, by enabling relatively cheap few-step sampling, can greatly improve performance. Specifically, we use a standard diffusion transformer (DiT) (Peebles & Xie, 2023) with an additional time embedding head. We also use a combination of data augmentation to enforce soft SO(3) (rotation) equivariance and subtraction of the mean to enforce translation invariance following Tan et al. (2025a;b).

**Formulations for $\mathcal{L}_{\text{avg}}$ in the context of Boltzmann Generators.** Many forms of $\mathcal{L}_{\text{avg}}$ have been explored in the context of fast generation (Geng et al., 2025a; Guo et al., 2025; Boffi et al., 2025a; Sabour et al., 2025); We discuss these losses in §B. In this work, we focus on the following loss, which is equivalent to the MeanFlow loss of Geng et al. (2025a), but also explore additive losses (Boffi et al., 2025a) similar to shortcut (Frans et al., 2025), split-MeanFlow (Guo et al., 2025). This choice is based on the superior performance of this loss in image experiments (Sabour et al., 2025), as well as its potential for efficient implementation, which we discuss below.

$$\mathcal{L}_{\text{avg}} \triangleq \mathbb{E}_{s,t,x_s}\left\|u_\theta(x_s, s, t) - \text{sg}\Big(v(x_s, s) - (t - s)\big(v(x_s, s)\partial_{x_s}u_\theta + \partial_s u_\theta\big)\Big)\right\|^2 \tag{10}$$

Note that since $x_s = sx_1 + (1 - s)x_0$, we can directly use $v(x_s, s) = x_1 - x_0$.

**Efficient Implementation.** As noted in multiple previous works, $\mathcal{L}_{\text{avg}}$ can be efficiently implemented using a single Jacobian vector product (JVP) call using forward automatic differentiation. Specifically, we have:

$$u_\theta(x_s, s, t), \frac{du_\theta}{ds} = \text{jvp}(u_\theta, (x_s, s, t), (v_s, 1, 0))$$

where the jvp function takes a callable function, inputs, and a vector (the vector part of the JVP).

Additionally, for this loss specifically, we can combine $\mathcal{L}_{cfm}$ and $\mathcal{L}_{avg}$ if we implement $v(x_s, s) = u_\theta(x_s, s, s)$, i.e. passing the same time to $u$, representing the instantaneous velocity. Specifically, we can implement the sum of the two losses by changing the distribution of $s, t$ in Eq. 10 to include some percentage of the time when $s = t$ to correspond with some fraction of $\mathcal{L}_{cfm}$ loss.

Our method is the first to require flow maps both in the forward and backward directions. We therefore need to consider the parameterization $u_\theta(x_s, t, s)$, i.e. the backwards direction, specifically at the discontinuity when $s = t$. When $t \to s^+$, then $u_\theta(x_s, s, t) = v(x_s, s)$, but when $t \to s^-$, then $u_\theta(x_s, s, t) = -v(x_s, s)$. To address this, we parameterize our flow map $u_\theta$ such that $u_\theta(x_s, s, t) = \text{sign}(t - s)h_\theta(x_s, s, t)$.

---

**Algorithm 1:** Training FALCON

**Input:** Sampleable $p_0$ and $p_1$, regularization weight $\lambda_r$, network $u_\theta$
**Output:** The trained network $u_\theta$

**while** *training* **do**
$\quad (x_0, x_1) \sim p_0 \times p_1$;
$\quad x_s \leftarrow sx_1 + (1 - s)x_0$;
$\quad v_s \leftarrow x_1 - x_0$;
$\quad u_\theta, \frac{\partial u_\theta}{\partial s} \leftarrow \text{jvp}(u_\theta, (x_s, s, t), (v_s, 1, 0))$;
$\quad u_{tgt} \leftarrow v_s - (t - s)\frac{\partial u_\theta}{\partial s}$;
$\quad \hat{x}_t \leftarrow x_s + (t - s)u_\theta$;
$\quad \hat{x}_s \leftarrow \hat{x}_t + (s - t)u_\theta(\hat{x}_t, t, s)$;
$\quad \mathcal{L}(\theta) \leftarrow \|u - \text{sg}(u_{tgt})\|^2 + \lambda_r\|x_s - \hat{x}_s\|^2$;
$\quad \theta \leftarrow \text{update}(\theta, \nabla_\theta\mathcal{L}(\theta))$;

**return** $u_\theta$;

---

## 4 EXPERIMENTS

In this section, we first demonstrate that FALCON achieves more scalable performance over state-of-the-art continuous flows across both global and local metrics on tri-alanine, alanine tetrapeptide, and hexa-alanine (Table 3). Next, we empirically demonstrate that FALCON flows exceed the performance of state-of-the-art discrete NFs, even when they are given vastly larger sampling budgets (Fig. 4). Then, we elucidate the importance of regularization in achieving invertibility and aiding generative performance (Fig. 6). Finally, we ablate inference schedules and show their impact on performance across metrics as a function of sampler choice (Fig. 7).

### 4.1 EXPERIMENTAL SETUP

**FALCON setup.** We evaluate FALCON with two losses, one based on MeanFlow (Geng et al., 2025a) (FALCON) as described in §3, and the other on an additivity loss (FALCON-A) (§B). We default to the first loss as it performed best or second best on all metrics on larger systems (Table 3).

**Datasets.** We evaluate the performance of FALCON on equilibrium conformation sampling tasks, focusing on alanine dipeptide (ALDP), tri-alanine (AL3), alanine tetrapeptide (AL4), and hexa-alanine (AL6). Datasets are obtained from implicit solvent molecular dynamics (MD) simulations with the `amber-14` force field, as detailed in §D.3. We train on biased data and test on a held-out unbiased dataset, using self-normalized importance sampling (SNIS) and force-field energies to compute log-likelihoods and re-sample molecules from the target Boltzmann density.

**Baselines.** We benchmark FALCON against both discrete and continuous normalizing flows. We include four discrete normalizing flow baselines: (**1**) SE(3)-EACF (Midgley et al., 2023); (**2**) RegFlow (Rehman et al., 2025); (**3**) SBG (Tan et al., 2025a) with standard SNIS (SBG IS); and (**4**) SBG with SMC sampling (SBG SMC), as well as three continuous flows: (**1**) ECNF (Klein et al., 2023); (**2**) ECNF++ (Tan et al., 2025a); and (**3**) BoltzNCE (Aggarwal et al., 2025), a recent SE(3)-equivariant architecture leveraging geometric vector perceptrons (GVPs) (Jing et al., 2021) on alanine dipeptide. For all continuous flows, samples and likelihoods are generated by integrating over the vector field using the Dormand–Prince 4(5) integrator with atol = rtol = $10^{-5}$ (Dormand & Prince, 1986) to ensure a fair comparison between methods. More details on architectures and parameters are covered in §D.1.

**Metrics.** We report Effective Sample Size (ESS), and the 2-Wasserstein distance on both the energy distribution ($\mathcal{E}$-$\mathcal{W}_2$), and dihedral angles ($\mathbb{T}$-$\mathcal{W}_2$). The full definitions of the metrics are included in §E. The energy captures local details, as minor atomic displacements yield large variations in the energy distribution, while $\mathbb{T}$-$\mathcal{W}_2$ captures global structure via mode coverage across metastable states. We

include energy histograms in the main text, with Ramachandran plots relegated to §F.4. For robustness, all quantitative experiments are performed on three seeds of the model and reported as mean $\pm$ standard deviation in the tables and figures. For all benchmarks, in cases where dashes are present, data was unavailable, except for SBG SMC (Tan et al., 2025a), where ESS is not a valid metric.

## 4.2 FALCON OUTPERFORMS STATE-OF-THE-ART METHODS

**Superior Scalability Over Continuous Flows.** Computing likelihoods in CNFs is computationally prohibitive, limiting their scalability in the Boltzmann Generator setting. Although the current state-of-the-art, ECNF++, performs exceptionally well on ESS and $\mathbb{T}\text{-}\mathcal{W}_2$ for alanine dipeptide (see Table 2) (Tan et al., 2025a), it fails to scale to larger molecules, as seen in Table 3. In contrast, for larger systems—tri-alanine, alanine tetrapeptide, and hexa-alanine—FALCON

Table 2: Results on alanine dipeptide. Best results are **bolded**, with second-best underlined.

| Algorithm ↓ | Alanine dipeptide (ALDP) | | |
| --- | --- | --- | --- |
| | ESS ↑ | $\mathcal{E}\text{-}\mathcal{W}_2 \downarrow$ | $\mathbb{T}\text{-}\mathcal{W}_2 \downarrow$ |
| BoltzNCE | — | $0.27 \pm 0.02$ | $0.57 \pm 0.00$ |
| SE(3)-EACF | $< 10^{-3}$ | $108.202$ | $2.867$ |
| RegFlow | $0.036$ | $0.519$ | $0.958$ |
| ECNF | $\underline{0.119}$ | $\underline{0.419}$ | $0.311$ |
| ECNF++ | $\mathbf{0.275 \pm 0.010}$ | $0.914 \pm 0.122$ | $\underline{0.189 \pm 0.019}$ |
| SBG IS | $0.030 \pm 0.012$ | $0.873 \pm 0.338$ | $0.439 \pm 0.129$ |
| SBG SMC | — | $0.741 \pm 0.189$ | $0.431 \pm 0.141$ |
| FALCON-A (Ours) | $0.097 \pm 0.007$ | $0.512 \pm 0.038$ | $\mathbf{0.180 \pm 0.005}$ |
| FALCON (Ours) | $0.067 \pm 0.013$ | $\mathbf{0.225 \pm 0.104}$ | $0.402 \pm 0.021$ |

substantially outperforms ECNF++ across all metrics, demonstrating superior scalability to larger molecular systems. The true MD energy distributions, learned proposals, and re-sampled energies for alanine dipeptide, tri-alanine, alanine tetrapeptide, and hexa-alanine are all shown in Fig. 3.

Table 3: Quantitative results on tri-alanine (AL3), alanine tetrapeptide (AL4), and hexa-alanine (AL6). Baseline methods presented with SNIS, unless stated otherwise. Evaluations are conducted over $2 \times 10^5$ samples.

| Algorithm ↓ | Tri-alanine (AL3) | | | Tetrapeptide (AL4) | | | Hexa-alanine (AL6) | | |
| --- | --- | --- | --- | --- | --- | --- | --- | --- | --- |
| | ESS ↑ | $\mathcal{E}\text{-}\mathcal{W}_2 \downarrow$ | $\mathbb{T}\text{-}\mathcal{W}_2 \downarrow$ | ESS ↑ | $\mathcal{E}\text{-}\mathcal{W}_2 \downarrow$ | $\mathbb{T}\text{-}\mathcal{W}_2 \downarrow$ | ESS ↑ | $\mathcal{E}\text{-}\mathcal{W}_2 \downarrow$ | $\mathbb{T}\text{-}\mathcal{W}_2 \downarrow$ |
| ECNF++ | $0.003 \pm 0.002$ | $2.206 \pm 0.813$ | $0.962 \pm 0.253$ | $0.016 \pm 0.001$ | $5.638 \pm 0.483$ | $1.002 \pm 0.061$ | $0.006 \pm 0.001$ | $10.668 \pm 0.285$ | $1.902 \pm 0.055$ |
| RegFlow | $0.029$ | $1.051$ | $1.612$ | $0.010$ | $6.277$ | $3.476$ | — | — | — |
| SBG IS | $0.052 \pm 0.013$ | $0.758 \pm 0.506$ | $0.502 \pm 0.016$ | $0.046 \pm 0.014$ | $1.068 \pm 0.495$ | $0.969 \pm 0.067$ | $0.034 \pm 0.015$ | $\underline{1.021 \pm 0.239}$ | $1.431 \pm 0.085$ |
| SBG SMC | — | $\underline{0.598 \pm 0.084}$ | $0.503 \pm 0.029$ | — | $\underline{1.007 \pm 0.382}$ | $1.039 \pm 0.069$ | — | $\underline{1.189 \pm 0.357}$ | $1.444 \pm 0.140$ |
| FALCON-A (Ours) | $\mathbf{0.104 \pm 0.004}$ | $1.385 \pm 0.182$ | $\mathbf{0.343 \pm 0.004}$ | $\mathbf{0.094 \pm 0.007}$ | $2.929 \pm 0.068$ | $1.094 \pm 0.034$ | $\mathbf{0.077 \pm 0.007}$ | $1.211 \pm 0.105$ | $\mathbf{1.163 \pm 0.112}$ |
| FALCON (Ours) | $0.077 \pm 0.004$ | $\mathbf{0.544 \pm 0.013}$ | $0.452 \pm 0.011$ | $0.055 \pm 0.003$ | $\mathbf{0.686 \pm 0.047}$ | $\mathbf{0.858 \pm 0.077}$ | $0.060 \pm 0.017$ | $\mathbf{0.892 \pm 0.311}$ | $1.256 \pm 0.132$ |

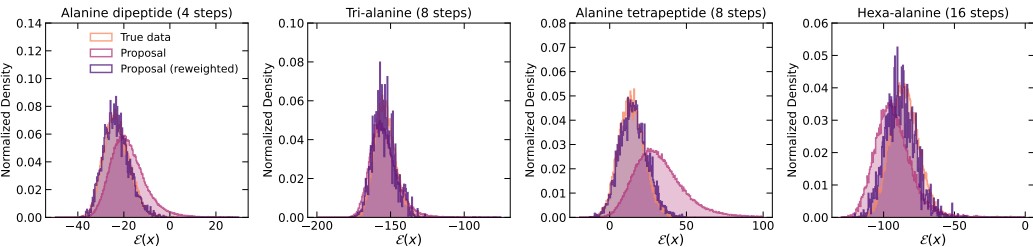

Figure 3: True MD energy distribution with best FALCON unweighted and re-sampled proposals for alanine dipeptide (**left**), tri-alanine (**center left**), and alanine tetrapeptide (**center right**), and hexa-alanine (**right**).

**Improved Sample Quality Over Discrete Flows.** Discrete NFs have recently been shown to be highly performant Boltzmann Generators (Rehman et al., 2025; Tan et al., 2025a;b). SBG (Tan et al., 2025a), based on the TARFlow architecture (Zhai et al., 2024), outperforms all previously reported methods across both global and local metrics (see Table 3). Here, we demonstrate that FALCON, even in a few steps, can outperform SBG across all metrics. We make two main claims to assert FALCON's competitive advantage in comparison with discrete NFs: (**1**) Discrete NFs, despite being fast one-step generators, consistently underperform compared to FALCON across all global and local metrics (Table 3); and (**2**) Increasing the number of samples can partially close this gap; however, even with $5 \times 10^6$ samples—250× more than those used to evaluate FALCON—SBG's performance on $\mathcal{E}\text{-}\mathcal{W}_2$ remains significantly worse than that of a 4-step FALCON Flow (as demonstrated in Fig. 4).

### 4.3 ANALYSIS OF COMPUTATIONAL EFFICIENCY

**Training Efficiency Compared to Discrete NFs.**
Discrete NFs benefit from fast inference, but are slow and unstable to train due to the maximum likelihood objective (Xu & Campbell, 2023; Andrade, 2024). By contrast, CNFs trained with a flow matching objective trade more stable and faster training for slower inference. When considering both training and evaluation time together (Table 4), we see that FALCON—despite being marginally slower at inference than the discrete NFs for the same number of samples—achieves faster cumulative training + inference times for superior performance due to the expedited training objective.

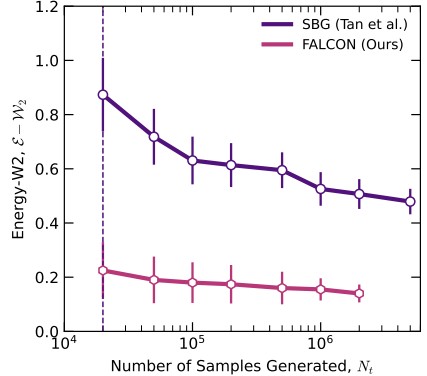

Figure 4: Performance with additional samples.

**Inference Efficiency Compared to CNFs.** A primary contribution of FALCON is the dramatic reduction in the computational cost required to achieve high-quality samples with accurate likelihoods. Fig. 2 directly illustrates this advantage. To reach a comparable level of performance on the $\mathbb{T}$-$\mathcal{W}_2$ metric for alanine dipeptide, a traditional CNF requires inference times that are two orders of magnitude longer than FALCON. This efficiency gain is what enables the use of more expressive alternative architectures and makes large-scale molecular sampling practical. For additional discussion, see §F.3.

Table 4: Cumulative training + inference time across flows. $2 \times 10^5$ samples evaluated with (atol = rtol = $10^{-5}$ for our CNF and 4-step FALCON). All experiments were conducted on a NVIDIA L40S with batch size 1024.

|  | ECNF++ | SBG | DiT CNF (Ours) | FALCON (Ours) |
|---|---|---|---|---|
| Alanine dipeptide | 12.52 | 16.83 | 9.56 | 7.65 |
| Tri-alanine | 19.59 | 24.67 | 17.54 | 11.45 |
| Alanine tetrapeptide | 32.17 | 41.67 | 24.10 | 18.84 |
| Hexa-alanine | 137.4 | 57.50 | 82.10 | 25.76 |

Table 5: Quantitative results on alanine dipeptide, tri-alanine, alanine tetrapeptide, and hexa-alanine compared to our Dopri5-integrated CNFs. Evaluations were conducted over $2 \times 10^5$ points across methods.

| System | Algorithm $\downarrow$ | ESS $\uparrow$ | $\mathcal{E}$-$\mathcal{W}_2$ $\downarrow$ | $\mathbb{T}$-$\mathcal{W}_2$ $\downarrow$ | NFE |
|---|---|---|---|---|---|
| Alanine dipeptide | FALCON-Dopri5 | $0.264 \pm 0.058$ | $0.442 \pm 0.048$ | $0.218 \pm 0.023$ | 257 |
|  | FALCON | $0.067 \pm 0.013$ | $0.225 \pm 0.104$ | $0.402 \pm 0.021$ | 4 |
| Tri-alanine | FALCON-Dopri5 | $0.125 \pm 0.034$ | $0.382 \pm 0.053$ | $0.370 \pm 0.093$ | 265 |
|  | FALCON | $0.077 \pm 0.004$ | $0.544 \pm 0.013$ | $0.452 \pm 0.011$ | 8 |
| Alanine tetrapeptide | FALCON-Dopri5 | $0.129 \pm 0.015$ | $0.665 \pm 0.047$ | $0.640 \pm 0.093$ | 200 |
|  | FALCON | $0.055 \pm 0.003$ | $0.686 \pm 0.047$ | $0.858 \pm 0.077$ | 8 |
| Hexa-alanine | FALCON-Dopri5 | $0.128 \pm 0.031$ | $1.013 \pm 0.115$ | $1.320 \pm 0.201$ | 207 |
|  | FALCON | $0.060 \pm 0.017$ | $0.892 \pm 0.311$ | $1.256 \pm 0.132$ | 16 |

**Inference vs. Accuracy Trade-off.** In FALCON, by using a flow map formulation, we can trade off performance for faster evaluation by adjusting the number of inference steps post-hoc. In Table 5, we show that a high-NFE, adaptive step solver achieves superior performance to the state-of-the-art continuous time ECNF++, as well as a few-step FALCON Flow; however, we also demonstrate that in this few-step regime, FALCON still outperforms every method considered (see Table 3) using two orders of magnitude fewer function evaluations (ranging between 4-16 steps, depending on the dataset). Depending on the compute budget and goal, we demonstrate in Fig. 5 how FALCON is able to interpolate between slow and accurate sampling with fast but less accurate sampling.

### 4.4 ABLATION STUDIES AND DESIGN CHOICES

**Verifying FALCON's Invertibility.** As CNFs are only invertible at convergence, we introduce a regularization term in the loss to promote numerical invertibility in the few-step regime. In Fig. 6,

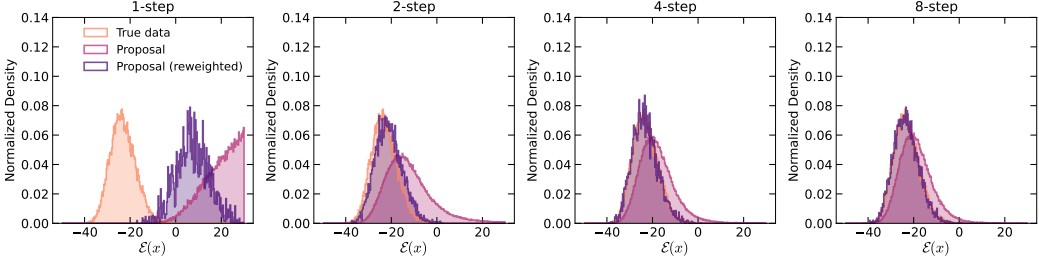

Figure 5: Improved proposal and re-weighted sample energies with increased steps for alanine dipeptide.

we demonstrate the trade-off from this term on both the ESS and 2-Wasserstein distance on dihedral angles: weak regularization leads to poor invertibility and degraded performance, whereas strong regularization enforces flow invertibility, albeit at the cost of reduced sample quality. We fix the regularization constant to $10.0$ for all experiments performed, unless stated otherwise to balance performance and numerical invertibility.

We also directly prove that an inverse exists for our trained flow, by training an auxiliary network to invert a frozen FALCON Flow in the forward direction. We find that FALCON achieves invertibility errors on the order of $10^{-4}$, which is the same order of magnitude as the invertibility of discrete and continuous NFs. Additional details can be found in §F.2 and Fig. 12.

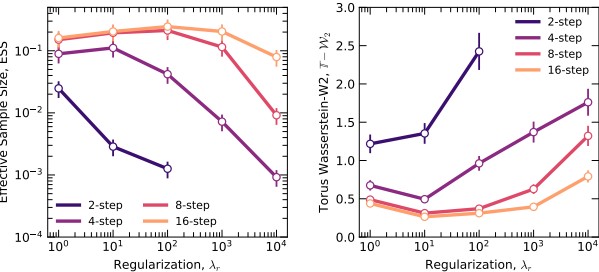

Figure 6: Performance trade-off with increasing regularization.

**Impact of Inference Schedules.** In the few step regime, performance can be significantly impacted by the choice of inference schedule. We run ablations on various schedules for alanine dipeptide with 8-steps, summarizing the results in Fig. 7. We note that EDM substantially outperforms all other schedulers, in agreement with observations from the diffusion literature (Karras et al., 2022); sampling more points near the data distribution proves beneficial in aiding generative performance as the variance of the flow field is higher closer to the target distribution. For all reported results, we use the EDM scheduler. In D.2.2, we provide additional details regarding scheduler definitions, inference setups, and parameter selection for EDM.

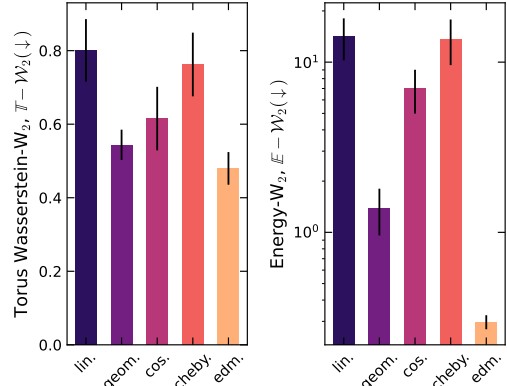

Figure 7: Performance vs. choice of inference schedule.

# 5 RELATED WORK

**Boltzmann Generators.** Boltzmann Generators (BGs) (Noé et al., 2019) are used to sample molecular conformations (Klein et al., 2023) and enable consistent estimates of thermodynamic observables (Wirnsberger et al., 2020; Rizzi et al., 2023; Schebek et al., 2024). While traditionally BGs are based on discrete normalizing flows, more recent work in machine learning makes use of more powerful continuous normalizing flow architectures for invariance (Köhler et al., 2020; Köhler et al., 2023) and expressive power (Klein et al., 2023; Klein & Noe, 2024). A few other works have explored the usage of approximate likelihoods (Draxler et al., 2024; Sorrenson et al., 2024; Aggarwal et al., 2025), but have until now been unable to scale. Rehman et al. (2025) also proposes a more

efficient BG using a new-regression-based objective to train discrete normalizing flow architectures, but the approach necessitates an invertible architecture limiting scalability and performance.

**Few-step Flows.** Diffusion and flow matching methods are now central to domains from vision (Song et al., 2021; Lipman et al., 2023) to scientific applications in material and drug discovery (Abramson et al., 2024; Noé et al., 2019). Scalable regression-based losses make these models fast to train, yet inference remains costly due to the need for numerous vector field integrations, motivating efforts to reduce computational expense. One- and few-step methods, like consistency models (Song et al., 2023; Song & Dhariwal, 2024; Geng et al., 2025b), shortcut models (Frans et al., 2025), and flow maps (Boffi et al., 2025a;b; Sabour et al., 2025; Guo et al., 2025) have gained recent attention. Concurrent work demonstrates that approximate likelihoods can be obtained through direct regression and are useful in guiding few step flows (Ai et al., 2025). Although there have been numerous efforts in improving generative performance in image settings, developing invertible few-step flows for scientific applications has seen substantially less interest. Our work, to the best of our knowledge, is the first demonstration of an invertible few-step flow with tractable likelihoods.

## 6 CONCLUSION

In this work, we introduced FEW-STEP ACCURATE LIKELIHOODS FOR CONTINUOUS FLOWS, a novel few-step flow-based generative model designed to address the long-standing challenge of scalable and efficient Boltzmann distribution sampling. Our approach successfully combines the expressiveness and training efficiency of modern flow-based models with a few-step sampling capability for fast yet accurate likelihood estimation. By leveraging a hybrid training objective, FALCON provides a practical solution for the computationally expensive likelihood evaluations that have historically limited the widespread adoption of Boltzmann Generators.

Our empirical results demonstrate that FALCON not only outperforms the existing state-of-the-art discrete normalizing flow models, but also provides a significant leap in computational efficiency over previous continuous flow models. We showed that our model is *two orders of magnitude faster* than an equivalently performing CNF-based Boltzmann Generator, making real-world, molecular sampling applications significantly more feasible. This represents a critical step toward unlocking the potential of Boltzmann Generators in fields ranging from drug discovery to materials science.

**Limitations.** Despite its advancements, FALCON has several key limitations that are crucial to acknowledge. First, while our results demonstrate that the computed likelihoods are *empirically good enough* for practical applications, we cannot efficiently guarantee their theoretical correctness. This represents a trade-off between computational efficiency and absolute theoretical certainty. Additionally, while theoretically possible, achieving true one-step generation remains out of reach for our current models, and we believe further architectural improvements and training methodologies are necessary to fully realize this potential.

Finally, our current research has primarily focused on the application of FALCON to Boltzmann Generation in molecular conformation sampling. Future work will explore applying our approach to Bayesian inference, robotics, and other domains where rapid and accurate likelihood estimation is critical. We also see potential in models with structured Jacobians (Rezende & Mohamed, 2015; Dinh et al., 2017; Zhai et al., 2024; Kolesnikov et al., 2025) to facilitate even faster sampling.

## ACKNOWLEDGEMENTS

DR received financial support from the Natural Sciences and Engineering Research Council's (NSERC) Banting Postdoctoral Fellowship under Funding Reference No. 198506. The authors acknowledge funding from UNIQUE, CIFAR, NSERC, Intel, and Samsung. The research was enabled in part by computational resources provided by the Digital Research Alliance of Canada (`https://alliancecan.ca`), Mila (`https://mila.quebec`), AITHYRA (`https://www.oeaw.ac.at/aithyra`), and NVIDIA.

## ETHICS STATEMENT

Our work is primarily focused on theoretical algorithmic development for faster and more accurate generative models for sampling from Boltzmann densities, with reduced focus on experimental implementation. However, we recommend that future users of our work exercise appropriate caution when applying it to domains that may involve sensitive considerations.

## REPRODUCIBILITY STATEMENT

We undertake multiple measures to ensure the reproducibility of our work. A dedicated section in §F.1 outlines the setup required to generate each of our reported figures. Further, we provide comprehensive information on the MD datasets used to train our models, including simulation parameters as well as the training, validation, and test splits used. We also include a separate section detailing model configurations, learning rate schedules, optimizer settings, hyperparameter choices, and other relevant aspects to facilitate reproduction in §D.1. We will also release all developed code publicly upon acceptance.

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

APPENDIX

## A  THEORY

**Proposition 1.** *Let $u_\theta^\star$ be a minimizer of Eq. 5 with respect to some $v$. Also, define the Jacobian of $X$ as $\mathbf{J}_X = \frac{\partial X}{\partial x_s}$, and the discrete flow map:*

$$X_u(x_s, s, t) = x_s + (t - s)u_\theta^\star(x_s, s, t) \tag{7}$$

*Then, for sufficiently smooth $u_\theta^\star$ and $v$ and for any $(s, t) \in [0, 1]^2$,*

1. *$X_u(\cdot, s, t)$ is an invertible map everywhere,*
2. *$\log p_t^{u^\star}(x_t) = \log p_s^{u^\star}(x_s) - \log |\det \mathbf{J}_{X_u}(x_s)|$ almost everywhere.*

We first define what we mean by sufficiently smooth in both propositions.

**Assumption 1.** *$u_\theta^\star$ is $C^1$ almost everywhere.*

This assumption on $u_\theta^\star$ allows the application of the change of variables formula almost everywhere, which is necessary to compute the log likelihood under $u$. We note that this is satisfied by most sufficiently expressive modern architectures including those using ReLU type activations, which are not $C^1$ everywhere, but are almost everywhere.

**Assumption 2.** *$v$ is uniformly Lipschitz continuous in $x$ and continuous in $t$.*

This assumption on $v$ is necessary in order to satisfy the Picard-Lindelöf Theorem. We note that if the Lipschitz condition does not hold then the Peano existence theorem implies that the initial value problem (IVP) of the ODE may not be invertible. If, however, we have both continuity in $x$ and $t$ as well as a Lipschitz condition for $v$ on $x$, then by the Picard-Lindelöf Theorem, we have existence and uniqueness of the IVP. We first recall Picard-Lindelöf.

**Theorem 1** (Picard-Lindelöf). *Let $D \subset \mathbb{R}^d \times \mathbb{R}$ be a closed rectangle with $(x_0, t_0) \in int\ D$, the interior of $D$. Let $v : D \to \mathbb{R}^d$ be a function that is continuous in $t$ and Lipschitz continuous in $x$ (with Lipschitz constant independent from $t$). Then there exists some $\epsilon > 0$ such that the initial value problem is:*

$$\frac{dx}{dt} = v(x_t, t), \quad x(t_0) = x_0 \tag{11}$$

*has a unique solution $x(t)$ on the interval $[t_0 - \epsilon, t_0 + \epsilon]$.*

Under these conditions on $v$ we are now able to prove the proposition.

*Proof.* Recall that if $u_\theta^\star$ minimizes Eq. 5:

$$\mathbb{E}_{s,t,x_s}\left[w(s,t)\|u_\theta(x_s, s, t) - \frac{1}{t-s}\int_s^t v(x_\tau, \tau)d\tau\|^2\right]$$

then we have that $u_\theta^\star(x_s, s, t) = \frac{1}{t-s}\int_s^t v(x_\tau, \tau)d\tau$ for all $s, t \in [0, 1]^2$. Furthermore we have

$$X_u(x_s, s, t) = x_s + (t - s)u_\theta^\star(x_s, s, t) \tag{12}$$

$$= x_s + (t - s)\frac{1}{t-s}\int_s^t v(x_\tau, \tau)d\tau \tag{13}$$

$$= x_s + \int_s^t v(x_\tau, \tau)d\tau \tag{14}$$

which by application of the Picard-Lindelöf theorem is invertible for all $s, t$, proving part 1 of the proposition.

To prove part 2, we note that Eq. 14 is differentiable with respect to $x_s$ and $X_s$ is invertible, therefore by the change-of-variables formula and Assumption 1, we arrive at part 2 of the proposition. $\square$

**Proposition 2.** *Let $u_\theta^\star$ be a minimizer of $\mathcal{L}_{inv}$ (Eq. (8)) with respect to some $v$. Then, for sufficiently smooth $u_\theta^\star$ and $v$ and for any $(s, t) \in [0, 1]^2$, $X_u(\cdot, s, t)$ is an invertible map everywhere, and $\log p_t^{u^\star}(x_t) = \log p_s^{u^\star}(x_s) - \log |\det \mathbf{J}_{X_u}(x_s)|$ almost everywhere.*

*Proof.* First we recall Eq. 8:

$$\mathcal{L}_{\text{inv}} = \mathbb{E}_{s,t,x_s} \|x_s - X_u(X_u(x_s, s, t), t, s)\|^2 \tag{15}$$

In this case, the minimizer of $\mathcal{L}_{\text{inv}}$ that is also in $C^1$ is pointwise invertible by definition as $X_u(\cdot, t, s)$ is the inverse of $X_u(\cdot, s, t)$ as $\mathcal{L}_{\text{inv}} \to 0$ everywhere.

For the change of variables to apply, we also need Assumption 1 to apply almost everywhere. $\quad\square$

We note that this loss ensures *pointwise* invertibility. Additional restrictions are needed to ensure that the log likelihood can be calculated efficiently.

## B  OTHER FORMULATIONS FOR $\mathcal{L}_{\text{AVG}}$

There are various formulations of $\mathcal{L}_{\text{avg}}$ that have been explored in the recent literature. We note that FALCON directly benefits from any new advancements in this rapidly-evolving research direction. Specifically, the following three different $\mathcal{L}_{\text{avg}}$ losses can be used for training flow maps.

1. $\mathcal{L}_1 \triangleq \mathbb{E}_{s,t,x_s} \left\| u_\theta(s, t, x_s) - \text{sg}\Big( v(x_s, s) - (t-s)\big(v(x_s, s)\partial_{x_s} u_\theta + \partial_s u_\theta\big)\Big) \right\|^2$, which is equivalent to the MeanFlow loss of Geng et al. (2025a), as well as the ESD objective in Boffi et al. (2025a). Note that since $x_s = sx_1 + (1-s)x_0$, we can directly use $v(x_s, s) = x_1 - x_0$.

2. $\mathcal{L}_2 \triangleq \mathbb{E}_{s,t,x_s} \left\| u_\theta(s, t, x_s) - \text{sg}\Big( \lambda u_\theta(x_s, s, r) + (1-\lambda)u_\theta(X_u(x_s, s, r), r, t)\Big) \right\|^2$, for $r = \lambda s + (1-\lambda)t$, is equivalent to the SplitMeanFlow loss of Guo et al. (2025), as well as the scaled PSD objective in Boffi et al. (2025a). In Boffi et al. (2025a), two ways of sampling the intermediate time $r$ are explored. Deterministic mid-point sampling, which sets $\lambda = 0.5$, as well as uniform sampling of $\lambda$. Guo et al. (2025) only explores uniform sampling. This is also a generalization of Frans et al. (2025), which additionally samples $s$ and $t$ in a binary-tree fashion. We denote this loss when integrated in our setting as FALCON-A. We use $\lambda = 0.5$ for all experiments in this setting as we found it was most stable.

3. $\mathcal{L}_3 \triangleq \mathbb{E}_{s,t,x_s} \left\| \partial_t X_u(x_s, s, t) - \text{sg}\Big( u_\theta\big(X_u(x_s, s, t), t, t\big)\Big) \right\|^2$ is the Lagrangian objective presented in Boffi et al. (2025a). We were not able to get this objective to train stably in our setting.

## C  EXTENDED RELATED WORK

### C.1  RELATION TO FREE-FORM FLOWS

Free-form Flows (FFF) enable arbitrary architectures to function as normalizing flows by jointly training an auxiliary network to approximate the inverse of the forward model (Draxler et al., 2024). The forward model maps data to latents, while the auxiliary network regularizes this mapping by learning a tractable estimator for the Jacobian term in the change of variables. As correctly highlighted by Draxler et al. (2024), although the loss encourages invertibility, this guarantee only holds when the reconstruction error is sufficiently small—a condition that can be difficult to meet in practice. In our setting, FALCON is trained with an entirely different loss function—albeit with a cycle-consistency term to promote invertibility—similar to that observed by Draxler et al. (2024). In addition, we explicitly validate invertibility for FALCON, in line with their observations through a dedicated experiment.

### C.2  RELATION TO REGFLOWS

RegFlows (Rehman et al., 2025) enable a more efficient training process for standard discrete normalizing flows by avoiding the maximum-likelihood (MLE) objective, whose unstable training dynamics often make optimization a challenge; instead, RegFlows distill the knowledge of a predefined invertible coupling—either using a pre-trained continuous normalizing flow or a pre-computed optimal

transport map—into a discrete normalizing flow via a regression objective. While this sidesteps issues associated with MLE and yields an efficient training pipeline, it also imposes a strict requirement on the existence of an invertible reference map, which must be provided in advance.

Further, RegFlows are constrained to inherently invertible architectures, which limits their expressivity and ultimately their scalability. As we show in Table 2 and Table 3, even when comparing against the strongest RegFlow variant (NSF), FALCON consistently outperforms it across all metrics, with pronounced gains on larger peptide systems. In contrast, FALCON is the first few-step Boltzmann Generator to employ a fully free-form architecture, surpassing state-of-the-art inherently invertible discrete flows (and even continuous normalizing flows) while avoiding the strict architectural constraints imposed by invertibility. Note, although FALCON is only guaranteed to be invertible either at convergence or via an additional regression objective, this relaxation enables substantially higher expressivity and delivers significant empirical performance gains. Lastly, we demonstrate the practical invertibility of FALCON through experiments performed in §F.2.

# D    EXPERIMENTAL DETAILS

All training experiments are run on a heterogeneous cluster of NVIDIA H100 and L40S GPUs using distributed data parallelism (DDP). All models were trained with three random seeds, and reported values are averages across runs. Additional training and inference details are included in the following. For benchmarks with existing methods, in cases where dashes are present, data was unavailable, except for the case with SBG SMC (Tan et al., 2025a), where ESS is not a valid metric.

## D.1    TRAINING DETAILS

**Architecture.** We adopt a Diffusion Transformer (DiT) backbone for FALCON, with the same model size used for all peptides. The details of the backbone configuration are in Table 6 below.

Table 6: Overview of FALCON configurations across datasets.

| Dataset | Hidden Size | Blocks | Heads | Cond. Dim. | Parameters (M) |
|---|---|---|---|---|---|
| Alanine dipeptide | 192 | 6 | 6 | 64 | 3.2 |
| Tri-alanine | 192 | 6 | 6 | 64 | 3.2 |
| Alanine tetrapeptide | 192 | 6 | 6 | 64 | 3.2 |
| Hexa-alanine | 192 | 6 | 6 | 64 | 3.2 |

**Training Configuration**    All models were trained with an exponential moving average (EMA) on the weights using a decay rate of 0.999. Logit values were clipped at 0.002, with compositional energy regularization disabled. For evaluation, we generated $10^4$ proposal samples, and used the same number for re-sampling and computing all metrics. Lastly, center-of-mass augmentation was applied with a standard deviation of $1/\sqrt{n}$, where $n$ is the number of particles.

**Hyperparameters**    Optimization was performed using AdamW with learning rate lr $= 5 \times 10^{-4}$, $\beta = (0.9, 0.999)$, $\epsilon = 10^{-8}$, and weight decay $10^{-4}$. A cosine annealing learning rate schedule with a warm-up phase covering 5% of the training iterations was also used.

## D.2    INFERENCE DETAILS

In this section we clarify the inference strategies used in this work. Both in terms of strategies used for likelihood computation, as well as how to schedule evaluations of FALCON.

### D.2.1    LIKELIHOOD COMPUTATION

Likelihood computation of CNFs generally uses the continuous-time formulation in Eq. (3), repeated below for clarity:

$$\begin{bmatrix} x_t \\ \log p_s^\theta(x_s) \end{bmatrix} = \int_0^s \begin{bmatrix} v_\theta(x_\tau, \tau) \\ -\mathrm{tr}\left(\frac{\partial v_\theta}{\partial x_\tau}\right) \end{bmatrix} d\tau, \text{ with initial condition } \begin{bmatrix} x_0 \\ \log p_0(x_0) \end{bmatrix}$$

There are two main degrees of freedom in approximating $\log p_1^\theta$ for some generated sample, the integrator, and the trace computation or approximation.

- **Integrator**: Following previous work we use the Dormand–Prince-45 (**Dopri5**) adaptive step size ODE integrator for all continuous-time models. To our knowledge, all Boltzmann Generators that use a base continuous normalizing flow generator use the Dopri5 solver as the importance sampling is otherwise too biased. Fixed-step size integrators, while effective for generation, are known to systematically overestimate likelihood for flow matching models.

- **Trace Computation**: There are two main strategies used in practice to estimate the trace $\text{tr}\left(\frac{\partial v_\theta}{\partial x_\tau}\right)$. Either using a direct exact computation, or using the Hutchinson trace approximation with a Monte–Carlo estimate of size 1. We find the exact computation necessary for Boltzmann Generator applications to maintain stability of the importance sampling.

In practice, this is implemented using a $d + 1$ dimensional ODE where the trace computation is computed in a vectorized manner using reverse mode automatic differentiation.

For FALCON, the continuous-time formulation no longer applies, and we need to use the exact change of variables formula at each discrete time-step as detailed in Algorithm 2.

---

**Algorithm 2:** Inference for FALCON

**Input:** Sampleable $p_0$, trained network $u_\theta$, time schedule $\{t_i\}_{i=0}^N$
**Output:** Samples $x_N \sim p_1^\theta$ with associated log likelihoods $p_N$
$x_0 \sim p_0$;
$p_0 \leftarrow \log p_0(x_0)$;
**for** $i \in [1, N]$ **do**
    $(s, t) \leftarrow (t_{i-1}, t_i)$;
    /* where $X_u(x_s, s, t) = x_s + (t - s)u_\theta(x_s, s, t)$                                 */
    $x_i \leftarrow X_u(x_{i-1}, s, t)$;
    $p_i \leftarrow p_{i-1} - \log |\det \mathbf{J}_{X_u}(x_{i-1})|$;
**return** $x_N, p_N$;

---

**Computational Cost**    Cost-wise, this computation turns out to be bottle-necked by function evaluations, and is requires approximately $T_l \cdot d$ function evaluations where $T_l$ is the number of integration-steps chosen by the adaptive step-size solver for the likelihood computation, and $d$ is the dimension of the system. This is in contrast to the $T_g$ function evaluations required to generate samples where $T_g$ is the number of integration steps chosen in this generation-only case. In practice, we find $T_l > T_g$, and often up to 2-3x larger.

FALCON on the other hand has a computational cost of $N \cdot d$ function evaluations where $N$ is the chosen number of steps to evaluate FALCON. In practice we set $N \ll T_l$, e.g. Table 5 where $T_l \approx 200$ and $N \approx 8$. We note that technically there is an additional $O(d^3)$ term to calculate the determinant. However, in practice this is not a bottleneck compared to the cost of function evaluations.

### D.2.2    INFERENCE SCHEDULERS

We evaluated five inference schedules to assess their impact on generative performance. The linear baseline distributes steps uniformly across the trajectory, while the geometric, cosine, Chebyshev, and EDM schedules bias step allocation to emphasize regions where the data distribution is more sensitive. The mathematical definitions and parameter settings for each schedule are provided below.

**Linear.**    The linear schedule spaces is distributed uniformly between 1 and 0, with $N \in \mathbb{N}$ denoting the total number of inference steps such that:

$$t_i = 1 - \frac{i}{N}, \quad i \in \{0, \dots, N\}. \tag{16}$$

**Geometric.** The geometric schedule exponentially allocates more resolution to early steps. We let $\alpha \in \mathbb{R}$ and $\alpha > 1$, denote the geometric base. For all experiments conducted, we set $\alpha = 2$:

$$t_i = \frac{\alpha^{N-i} - 1}{\alpha^N - 1}, \quad i \in \{0, \ldots, N\}. \tag{17}$$

**Cosine.** The cosine schedule follows a squared-cosine law, concentrating more steps near $t = 0$. This has been used in diffusion models for improved stability. With $N \in \mathbb{N}$:

$$t_i = \cos^2\left(\frac{\pi}{2} \cdot \frac{i}{N}\right), \quad i \in \{0, \ldots, N\}, \quad t_N = 0. \tag{18}$$

**Chebyshev.** The Chebyshev schedule, derived from the nodes of Chebyshev polynomials of the first kind, minimizes polynomial interpolation error, and distributes steps more densely near boundaries:

$$t_i = \frac{1}{2}\left(\cos\left(\frac{\pi(i+0.5)}{N+1}\right) + 1\right), \quad i \in \{0, \ldots, N\}, \quad t_N = 0. \tag{19}$$

**EDM.** The EDM schedule (Karras et al., 2022) parameterizes the noise level $\sigma$ using a power-law with exponent $\rho \in \mathbb{R}^+$. The interpolation is performed in $\rho$-space to allocate more steps where the generative process is most sensitive. For all experiments performed, we use the same parameters proposed by Karras et al. (2022), with $\rho = 7$, $\sigma_{\min} = 10^{-3}$, and $\sigma_{\max} = 1$, and $\sigma_{\min}, \sigma_{\max} \in \mathbb{R}^+$:

$$\sigma_i = \left(\sigma_{\max}^{1/\rho} + \frac{i}{N}\left(\sigma_{\min}^{1/\rho} - \sigma_{\max}^{1/\rho}\right)\right)^\rho, \quad i \in \{0, \ldots, N\}, \tag{20}$$

$$t_i = \frac{\sigma_i - \sigma_{\min}}{\sigma_{\max} - \sigma_{\min}}, \quad t_N = 0. \tag{21}$$

This formulation ensures a smooth transition between $\sigma_{\max}$ (high noise) and $\sigma_{\min}$ (low noise, near-deterministic refinement).

### D.3 DATASET DETAILS

For the datasets used, we follow the same training, validation, and test split used by (Tan et al., 2025a), where a single MCMC chain is decomposed into $10^5$, $2 \times 10^4$, and $10^4$ samples for training, validation, and test. The training and validation data are each taken as contiguous regions of the chain to simulate the realistic scenario where you have generated an MCMC trajectory and would like to use a Boltzmann Generator to continue generating samples. The data is split so that (after warmup) the first $10^5$ samples are for train, the next $2 \times 10^4$ are for validation, and the test samples are uniformly strided sub-samples from the remaining MCMC chain. Earlier parts of the trajectory undersample certain modes enabling a biased training set that we attempt to debias through access to the energy function and SNIS. MD simulations were all run for 1 µs, with a timestep of 1 fs, at temperatures of 300K, 310K, and 300K for alanine dipeptide, tri-alanine, and alanine tetrapeptide, respectively, in line with those conducted by Klein & Noe (2024). The force fields used for all three molecules, in the same order, were the Amber ff99SBildn, Amber 14, and Amber ff99SBildn, in line with prior work by Tan et al. (2025a).

**Alanine Dipeptide.** The Ramachandran plots for the training and test split are provided in Fig. 8. Following Klein et al. (2023), we up sample the lowest sampled mode (middle right) to make the problem easier for BGs. This also has the added benefit of testing what happens to FALCON when the training dataset is biased relative to the true distribution.

**Tri-alanine.** Similarly to alanine dipeptide, the Ramachandran plots for the training and test split are in Fig. 9. As there are two pairs of torsion angles that parameterize the system, there are two sets of Ramachandran plots included for each training and test. In tri-alanine, we can see that the training set actually misses a mode entirely ($\psi_1 \approx \pi/3$), and undersamples this mode for $\psi_2$ relative to the test set. This is a great test of finding a new mode (in $\psi_1$) and correctly weighting a mode (in $\psi_2$).

**Alanine Tetrapeptide.** For the tetrapeptide, there are three sets of Ramachandran plots each for training and test given the three pairs of torsions angles that parameterize the molecule, in Fig. 10.

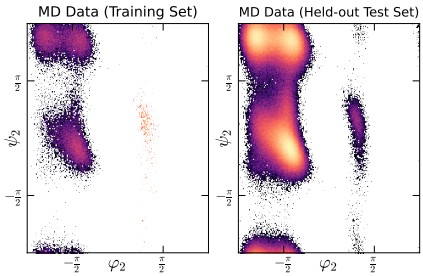

Figure 8: **Left:** Training data for alanine dipeptide; **Right:** Test data for alanine dipeptide.

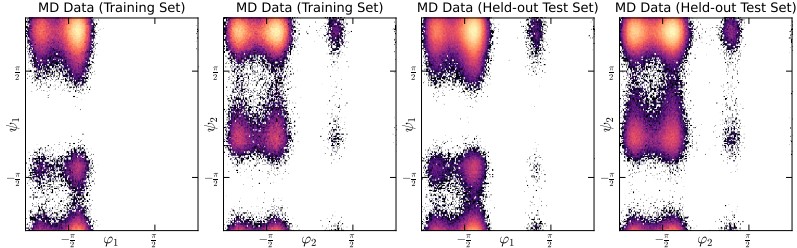

Figure 9: **Left and left center:** Training data for tri-alanine; **Right center and right:** Test data for tri-alanine.

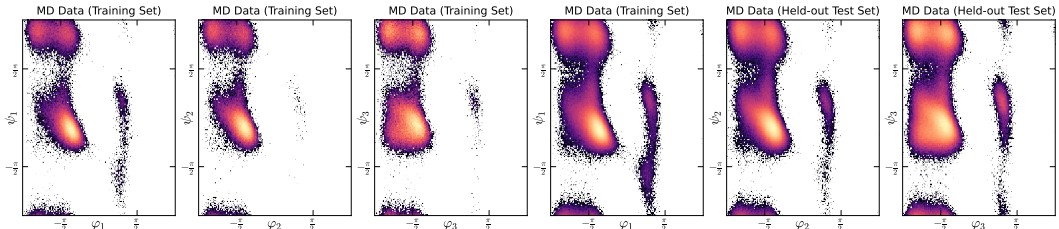

Figure 10: **First three:** Training data for alanine tetrapeptide; **Last three:** Test data for alanine tetrapeptide.

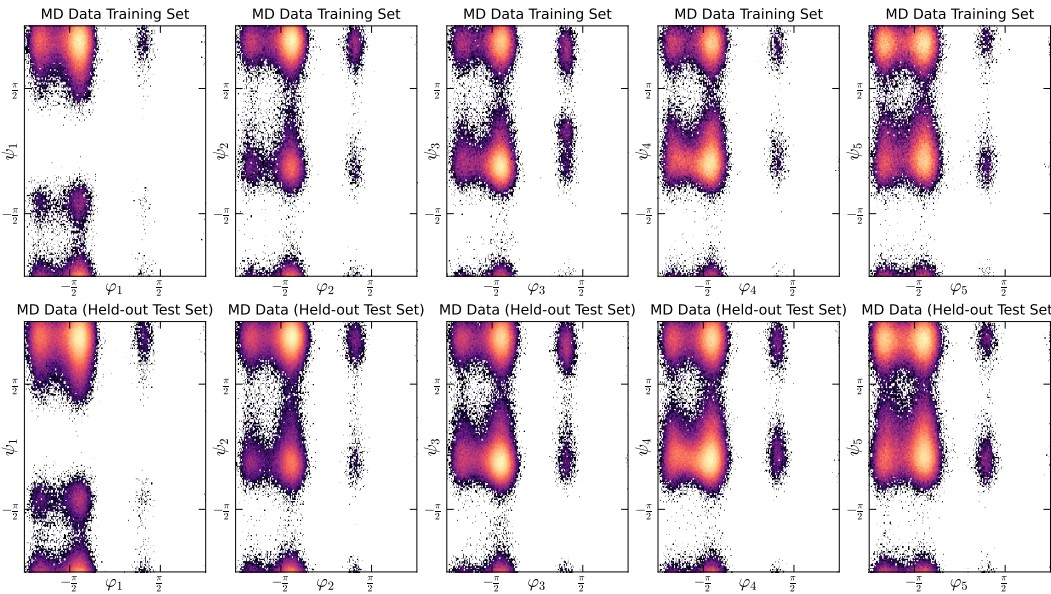

Figure 11: **First five:** Training data for hexa-alanine; **Last three:** Test data for hexa-alanine.

**Hexa-alanine.** For hexa-alanine, we also include Ramachandran plots in Fig. 11. The first row shows training data, while the second row shows a held-out test set used to evaluate performance.

# E    METRICS

**Effective Sample Size (ESS).** To quantify sampling efficiency, we compute the effective sample size (ESS) following Kish's definition (Kish, 1957). Given $N \in \mathbb{N}$ generated particles with unnormalized importance weights $\{w_i\}_{i=1}^N \subset \mathbb{R}^+$, the ESS is normalized by $N$ as:

$$\text{ESS}(\{w_i\}_{i=1}^N) \triangleq \frac{1}{N} \frac{1}{\sum_{i=1}^N w_i^2} \left( \sum_{i=1}^N w_i \right)^2. \tag{22}$$

The ESS reflects how many independent, equally-weighted samples would provide equivalent statistical power to the weighted sample. Higher ESS is desirable for more performant models.

**2-Wasserstein Energy Distance ($\mathcal{E}$-$\mathcal{W}_2$).** To compare energy distributions, we compute the 2-Wasserstein distance between generated and reference samples from our MD dataset. For distributions $p, q \in \mathcal{P}(\mathbb{R})$ over energy values, and $\Pi(p,q)$ the set of admissible couplings, we define:

$$\mathcal{E}\text{-}\mathcal{W}_2(p,q)^2 \triangleq \min_{\pi \in \Pi(p,q)} \int_{\mathbb{R} \times \mathbb{R}} |x - y|^2 \, d\pi(x,y). \tag{23}$$

The $\mathcal{E}$-$\mathcal{W}_2$ measures how closely the generated energy histogram aligns with that of the reference, with high sensitivity to structural accuracy due to bond-length–dependent energies. Since small perturbations in local structure induce large fluctuations in the energy distribution, this metric captures this variance, with lower values of $\mathcal{E}$-$\mathcal{W}_2$ being more favourable for generative models.

**Torus 2-Wasserstein Distance ($\mathbb{T}$-$\mathcal{W}_2$).** To assess structural similarity in torsional space, we compute a 2-Wasserstein distance over dihedral angles. For a molecule with $L \in \mathbb{N}$ residues, define the dihedral vector as:

$$\text{Dihedrals}(x) = (\phi_1, \psi_1, \dots, \phi_{L-1}, \psi_{L-1}) \in [0, 2\pi)^{2(L-1)}. \tag{24}$$

The cost on the torus accounts for periodicity of angles:

$$c_{\mathcal{T}}(x,y)^2 = \sum_{i=1}^{2(L-1)} \left[ (\text{Dihedrals}(x)_i - \text{Dihedrals}(y)_i + \pi) \bmod 2\pi - \pi \right]^2. \tag{25}$$

The torus 2-Wasserstein distance between two distributions $p, q \in \mathcal{P}([0, 2\pi)^{2(L-1)})$ is then:

$$\mathbb{T}\text{-}\mathcal{W}_2(p,q)^2 \triangleq \min_{\pi \in \Pi(p,q)} \int c_{\mathcal{T}}(x,y)^2 \, d\pi(x,y). \tag{26}$$

This captures macroscopic conformational differences while respecting angular periodicity. The $\mathbb{T}$-$\mathcal{W}_2$ provides a more global assessment of performance, for instance revealing missed conformational modes that would not be captured by the $\mathcal{E}$-$\mathcal{W}_2$.

# F    ADDITIONAL EXPERIMENTS

## F.1    DETAILS FOR FIGURE GENERATION

In Fig. 2, we compare performance on the $\mathbb{T}$-$\mathcal{W}_2$ metric for alanine dipeptide between FALCON and our continuous-time DiT CNF. For FALCON, accuracy is controlled by the number of inference steps (1–8). For the DiT CNF, we consider three settings: (**1**) adaptive step Dormand–Prince 4(5) with exact Jacobian trace evaluation, varying atol/rtol from $10^{-1}$ to $10^{-5}$; (**2**) the same tolerances but instead using the Hutchinson trace estimator, trading performance for faster likelihoods with higher variance; and (**3**) fixed-step Euler integration with step sizes from 4 to 256 with exact Jacobian traces, where the upper bound is chosen to roughly match the number of function evaluations needed for Dopri5. The models and training configurations used are presented in §D.1.

In Fig. 3, we demonstrate the energy distributions for unweighted and re-weighted samples for our most performant FALCON Flows. For alanine dipeptide, tri-alanine, alanine tetrapeptide, and hexa-alanine, 4, 8, 8, and 16 steps were used for figure generation. Energy distributions reveal microscopic detail, as marginal changes in local atomic position can have significant impacts on total energy. The best models for each molecular system were used, with pertinent details on model size, training configurations, and hyperparameters detailed in §D.1. Similarly, in Fig. 5, we show the proposal and re-weighted sample estimates for alanine dipeptide, and demonstrate how an increasing number of steps, improves the energy distribution.

In Fig. 4, we compare the performance between SBG's discrete NFs and FALCON, illustrating that despite more samples increasing performance across metrics, the approach is still unable to reach FALCON's performance. For SBG, we specifically take their best model weights for the TarFlow architecture from: `https://github.com/transferable-samplers/transferable-samplers`, and draw $N_s \in \{10^4, 2 \times 10^4, 5 \times 10^4, 10^5, 2 \times 10^5, 5 \times 10^5, 10^6, 2 \times 10^6, 5 \times 10^6\}$ samples three different times (the error bars are representative of the three draws) and evaluate $\mathcal{E}\text{-}\mathcal{W}_2$ for each set.

In Fig. 6, we investigate how the strength of regularization impacts performance on ESS and $\mathcal{E}\text{-}\mathcal{W}_2$. Specifically, we demonstrate that small amounts of regularization enable generative performance but impede invertibility, while too much regularization detrimentally impacts sample quality. This trade-off leads to an optimum on both metrics. We investigate $\lambda_r \in \{10^0, 10^1, 10^2, 10^3, 10^4, 10^5\}$, and conclude upon $\lambda_r = 10^1$. To ascertain the optimal $\lambda_r$, we ran these experiments on alanine dipeptide, using the same model details and configurations highlighted in §D.1.

Fig. 7 demonstrates the sensitivity the inference schedule plays on generative performance. For this figure, we took our best models—trained with the details presented in §D.1—and ran inference using all three trained seeds to generate uncertainties for each inference schedule. We used our 8-step system, as inference scheduler choice is less important the fewer steps are used.

## F.2 PROOF OF INVERTIBILITY

In our loss, we use a cycle-consistency term that regularizes training to promote numerical invertibility. We see in Fig. 6, the introduction of this modified loss aids generative performance; however, the forward-backward reconstruction yields errors on the order of $10^{-2}$, indicating an approximately, but not entirely invertible model. To prove that the flow is invertible (but the inverse is challenging to discover during training), we train an auxiliary FALCON Flow in the reverse direction. We use the same network to parameterize the reverse flow as described in §D.1, with the same training configuration and hyperparameter set. Next, we freeze the forward model (which goes from latents $\rightarrow$ data), generate synthetic data ($2 \times 10^5$ prior-target sample pairs) and train the auxiliary model on these reflow targets to learn the mapping from data back to latents. In Fig. 12, we illustrate the loss curve of the trained auxiliary FALCON. To

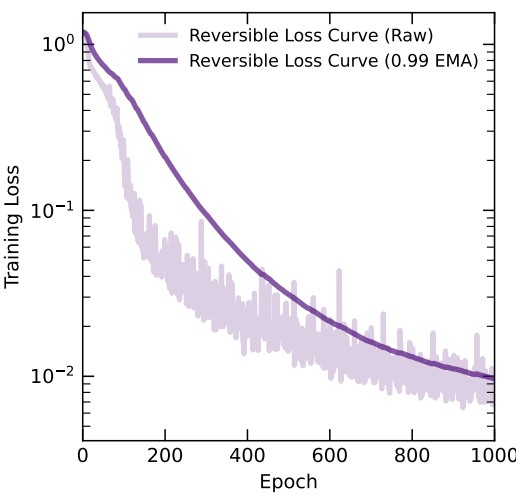

Figure 12: Auxiliary model loss from reflow target training on forward flow.

evaluate invertibility, we draw i.i.d. samples from our prior distribution, pass them through the frozen forward model, and test the auxiliary reverse model on these unseen latents. We evaluate an $\ell_2$ error on the recovered latents to find reconstruction within $10^{-4}$ after 1000 epochs matching the reconstruction accuracy of discrete NFs that are invertible by design. These results support the claim that the learned flow is indeed invertible.

### F.3 PERFORMANCE AGAINST NUMBER OF FUNCTION EVALUATIONS

Fig. 13 highlights the efficiency of FALCON in terms of number of function evaluations. We see that FALCON achieves the same torus 2-Wasserstein performance as our DiT CNF with Dopri5, while requiring over two orders of magnitude fewer function evaluations. Whereas Fig. 2 quantified efficiency in terms of inference time, here we measure the number of function evaluations against both fixed-step solvers (Euler with 4–256 steps) and adaptive solvers (Dopri5 run until reaching target tolerances atol = rtol $\in$ $\{10^{-2}, 10^{-3}, 10^{-4}, 10^{-5}\}$ for Hutchinson and $\in$ $\{10^{-2}, 10^{-3}, 10^{-4}, 10^{-5}, 10^{-6}\}$ for exact).

Although Hutchinson's trace estimator yields a speedup relative to exact Jacobian computation for CNFs, as seen in Fig. 2, the number of function evaluations needed is higher than the exact Jacobian computation. In either setting, however, FALCON

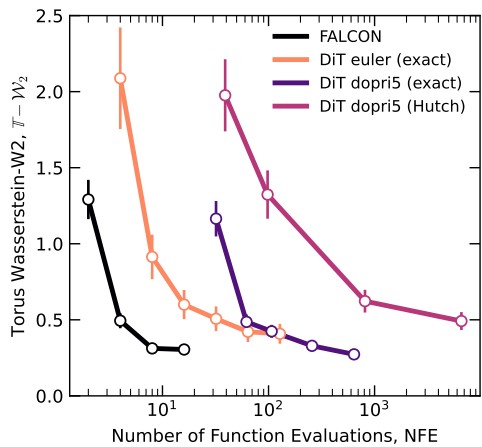

Figure 13: Performance of FALCON vs. our DiT CNFs as a function of NFEs.

still remains substantially faster. Even with a 4-step solver, FALCON matches the accuracy of DiT CNFs using Hutchinson at atol = rtol = $10^{-5}$, with an 8-step solver nearly matching the performance of a DiT CNF with Dopri5 set to an atol = rtol = $10^{-6}$.

### F.4 RAMACHANDRAN PLOTS

**Alanine Dipeptide.** We demonstrate FALCON's capacity to learning global features through the Ramachandran plots in Fig. 14 for alanine dipeptide. We include both the held-out test set and the learned model's map, showcasing its ability to debias the data and capture the true MD distribution. Specifically, the undersampled $\phi$ mode in the training data is correctly upweighted in the learned model's predictions, indicating accurate likelihood estimates from the learned flow.

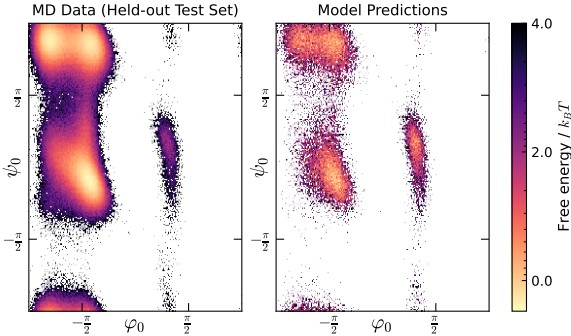

Figure 14: **Left:** Test data for alanine dipeptide; **Right:** FALCON's angular predictions for alanine dipeptide.

**Tri-alanine.** Similarly, we show the Ramachandran plots for tri-alanine in Fig. 15 exhibiting similar behaviour. Most conformations are correctly captured, with some modes being underweighted.

**Alanine Tetrapeptide.** Next, we show the Ramachandran plots for alanine tetrapeptide in Fig. 16.

**Hexa-alanine.** Finally, we show the Ramachandran plots for hexa-alanine in Fig. 17.

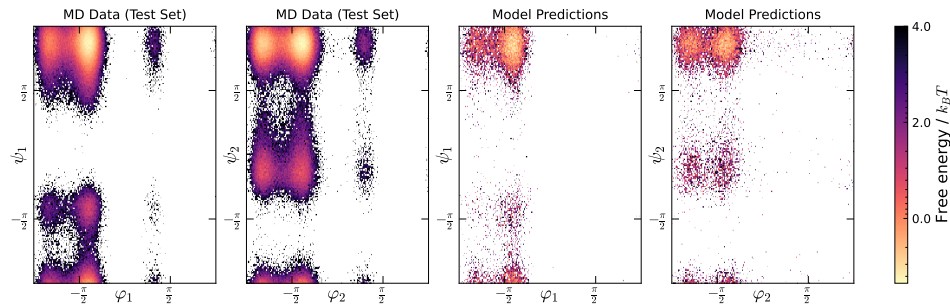

Figure 15: **Left and left center:** Test data for tri-alanine; **Right and right center:** FALCON's angular predictions for tri-alanine.

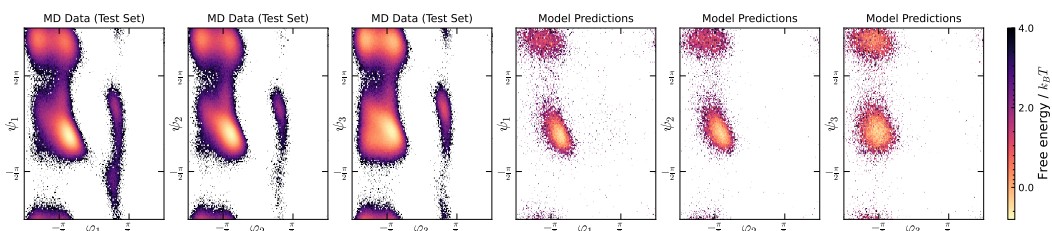

Figure 16: **First three:** Test data for alanine tetrapeptide; **Last three:** FALCON's angular predictions for alanine tetrapeptide.

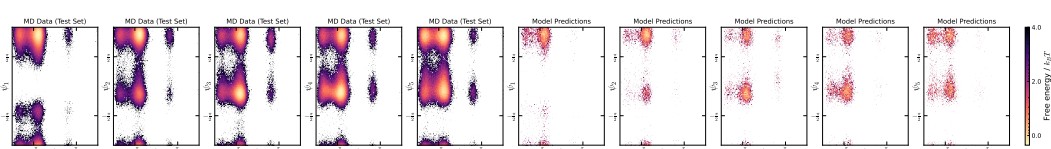

Figure 17: **First five:** Test data for hexa-alanine; **Last five:** FALCON's angular predictions for hexa-alanine.

