# OpenReview forum: "FALCON: Few-step Accurate Likelihoods for Continuous Flows"
_ICLR.cc/2026/Conference — ICLR 2026 Oral_

### Official Review · Reviewer_tZEt · 2025-10-19

**Soundness:** 3
**Presentation:** 3
**Contribution:** 3
**Rating:** 6
**Confidence:** 4

**Summary:**

This paper proposes to train a mean-flow model with an additional soft constraint on the flow invertibility by optimising the $\mathcal{L}_{inv}$ (Equation 8). Once the optimum of this inv loss is achieved, the log-prob of a generated sample can be calculated by change-of-variable. The proposed model, FALCON, could generate samples with few network calls while having access to model density, which can be applied as a Boltzmann generator.

The method is clearly written, and the empirical results show a clear gap between other baselines on either ESS or sample quality related metrics when scaling to complex tasks. However, the main concern lies in the low ESS of FALCON on ALDP, which is significantly lower than one of the baseline ECNF++.

Overall, the reviewer recommends a weak accept.

**Strengths:**

1. The method is clearly written

2. The proposed method is simple and effective

3. The experiments show the scalability of the proposed method, which outperforms other baselines

**Weaknesses:**

The main weakness of the paper lies in the experimental result on ALDP, where the ESS of FALCON is much lower than that of ECNF++. It would be great to elaborate more on why this happens. On the other hand, to showcase the effectiveness of proposed method, one could use the same network architecture as ECNF++ and then train FALCON. Such setting would be fairer to illustrate the effectiveness of the invertibility loss.

On the other hand, it is not clear how invertible the trained network is. In particular, the author could elaborate more on the invertibility of $X(x, s, t)$ by using a 2-dimensinonal heatmap. Intuitively speaking, for farer s and t, the invertibility is more difficult to maintain, and therefore increasing the number of NFE could improve ESS.

**Questions:**

Please see weakness.

---

> ### Author Response · Authors · 2025-11-21
> **Response to Reviewer tZEt**
>
> We would like to thank the reviewer for their thorough review and insightful comments on our work. We address all their points in the following.
>
> 1. **FALCON ESS performance compared to ECNF++ on ALDP** The reviewer correctly identified that FALCON’s ESS on ALDP was lower than ECNF++. This arises primarily from the integration scheme, not the modelling capability or loss. We elaborate further below and have added additional discussion in the manuscript.
> - **Comparison to few-step generators**: When comparing FALCON vs. the best 1-step generator (SBG) we find FALCON has roughly 2x the ESS.
> - **Control over efficiency-performance tradeoff**: FALCON gives users the flexibility to trade inference speed for higher ESS/accuracy, unlike ECNF++, which requires expensive Dopri5 integration. We believe this tradeoff is a strength of FALCON rather than a weakness.
> - **Number of integration steps**: The ECNF++ baseline uses a Dopri5 integrator (~250 function evaluations). Our reported FALCON results used a fast 2-step integrator.
> - **Apples-to-Apples Comparison**: When we evaluate FALCON using the same Dopri5 integrator (see Tab. 5), FALCON nearly matches the ECNF++ on ESS and Torus while vastly outperforming on Energy.
> - **Speed Advantage**: Crucially, with the same Dopri5 integrator, FALCON is still **~3x faster** than ECNF++ due to its efficient architecture during inference.
> - **Scale Advantage**: On all systems larger than ALDP, on which ECNF++ was optimized, FALCON outperforms ECNF++ even with many fewer steps.
>
> 2. **Architecture comparison (ECNF++ backbone)** Regarding the suggestion to train FALCON using the ECNF++ architecture:
> A core contribution of FALCON is enabling the use of free-form architectures (like DiTs or MLPs) rather than being constrained to specialized, expensive equivariant layers required by ECNF++ or SBG.
> While applying the FALCON loss to the ECNF++ architecture is theoretically possible, it would reintroduce the computational bottlenecks we aim to solve. Our results show that a standard DiT trained with FALCON outperforms the specialized ECNF++ architecture on larger systems (Tab. 3) and on ALDP with a similar number of steps (Tab. 5), validating our "free-form" approach.
> Tab. 5 also shows that FALCON with fewer steps is unlikely to outperform FALCON with more steps. Therefore we do not anticipate an equivariant architecture trained with FALCON to outperform our current free-form architecture.
>
> 3. **Empirical Invertibility** Lastly, the reviewer very correctly points out the importance of guaranteeing invertibility. There are two components of this question that we answer separately:
> - **NFE vs. Invertibility** The reviewer alludes to how increasing NFE can improve ESS (less spacing between $s$ and $t$). This is correct and can be seen in two cases. First, we see from Tab. 5 that going from a few-step FALCON to Dopri5 FALCON always leads to improvements in ESS across all peptide systems. This is in line with the reviewers' points about how increasing the accuracy of the integrators leads to an intrinsically more invertible system, and consequently higher ESS (and improved performance on other metrics). Secondly, in Fig. 6, we see how the invertibility regularization parameter impacts performance on ESS and the Torus metric. Specifically, we see that as we go from 2-steps to 16-steps, there is a clear increase in ESS and drop in Torus, as expected, validating that more steps leads to better performance.
> - **Existence vs. construction of inverse function** It is unfortunately difficult and expensive to make an $s$, $t$ heatmap of invertibility because we don’t have access to the exact inverse. As discussed in Appendix F.2, and the next point, we note that this requires specific training of the inverse.
> - **Quantitative verification (trained reconstruction)**: To prove invertibility empirically—albeit with an inverse that is challenging to find during training—we trained an auxiliary FALCON model on reflow targets generated from the frozen forward model, yet to go backwards (in other words, the forward model operates on latents $\to$ data, while the reverse model goes from data $\to$ latents). Then, we sampled a random set of latents, passed it through the frozen forward model, and inverted it using the newly-trained auxiliary FALCON, demonstrating recovered latents to within 1e-4. This was in line with reconstruction errors observed with discrete NFs—architectures that are invertible by design— conclusively proving that our trained model is indeed invertible. This procedure is detailed in Appendix F.2. Overall, as the reviewer correctly points out, invertibility of these models is essential to ensure reliable likelihoods for importance sampling, and here, we clearly demonstrate that through the additional experiment.
>
> We hope these clarifications regarding the integration scheme and architectural choices address the reviewer's concerns.

---

> ### Comment · Reviewer_tZEt · 2025-11-26
> **Response to author**
>
> The reviewer would like to firstly thank the authors for delivering the detailed responses that trying to answer the reviewer's concerns and most of the reviewer's concerns are solved.
>
> However, as the invertibility is approximated, the reviewer is not convinced that in the middle generation-step regime, if the errors would be accumulated. The author shows a "scaling law" of the num-of-step in Figure 6, and also FALCON-Dorip5 in table 5. The first case is up to 16 steps. While the latter case doesn't indicate too much information about the issues introduced by invertibility as the discretization is relatively large where each step can already be consider "invertible". The reviewer is curious on, when increasing the num-of-steps while not reaching the size that the invertibility of the network itself is ignorable, if the performance would be degenerated as errors of the  invertibility of network could accumulate.
>
> Overall, the reviewer appreciates the authors’ responses and would like to raise the score by one point. However, because the scoring scale does not include 7, the reviewer will retain the current score of 6 and flag this to the AC.

---

### Official Review · Reviewer_vNfw · 2025-10-29

**Soundness:** 4
**Presentation:** 4
**Contribution:** 4
**Rating:** 10
**Confidence:** 4

**Summary:**

This paper introduces FALCON, a framework designed to overcome a long-standing bottleneck of flow matching and related continuous-time generative models: although training is efficient, evaluating accurate likelihoods remains computationally expensive due to the need to integrate the divergence term along the ODE trajectory.

FALCON proposes a theoretically grounded and computationally efficient approach that enables few-step yet accurate log-likelihood estimation by optimizing a surrogate network to approximate the evolution of the log-density under flow dynamics. The method provides an effective approximation that avoids explicit computation of the ODE while maintaining high accuracy in the estimated likelihoods. Theoretical analyses support its consistency and error bounds, and extensive experiments demonstrate that FALCON achieves comparable or better accuracy than full integration methods at a fraction of the cost.

Overall, the paper offers a mathematically elegant and practically useful solution to a central problem in flow-based models.

**Strengths:**

1. The paper addresses a core unresolved issue in the flow-matching and continuous normalizing flow (CNF) literature — accurate and efficient likelihood computation — using a very novel and conceptually clean idea. The formulation of few-step likelihood estimation through continuous optimization is both original and impactful.

2. The theoretical development is rigorous, with solid mathematical grounding that explains why the surrogate optimization scheme yields unbiased or low-bias likelihood estimates.

3. Empirically, the experiments are thorough and convincing, covering a range of flow architectures and datasets. The results consistently demonstrate that FALCON significantly reduces computational overhead without sacrificing estimation accuracy.

In summary, this contribution is potentially highly influential for both the theoretical and applied machine-learning communities. It removes a practical barrier that has limited the deployment of flow-based models in real-world density estimation. FALCON may become a standard component for efficient probabilistic inference with continuous flows.

**Weaknesses:**

Overall, the work is strong and technically solid, but a few aspects could benefit from further clarification or discussion:

1. Since the method still depends implicitly on the invertibility and numerical conditioning of the local Jacobian, it would be important to analyze the behavior when the Jacobian determinant approaches zero, i.e., when the approximate mean velocity field u_theta becomes locally near-singular. Could such regions lead to unstable or biased likelihood estimates? Most practical architectures cannot guarantee strict invertibility.

2. Theoretical analysis likely assumes certain smoothness properties of the vector field to ensure the validity of the continuous approximation. These assumptions should be clearly stated and their necessity discussed, especially for large-scale neural networks that are not strictly smooth.

**Questions:**

1. If the Jacobian determinant becomes close to zero in some regions (i.e., the mapping is nearly non-invertible), how does this affect the numerical stability and bias of the estimated likelihood? Can the authors provide practical guidance or regularization strategies to mitigate this issue?

2. What specific smoothness or Lipschitz continuity assumptions are required on u_theta for the theoretical guarantees to hold? Would the results still apply to models with ReLU-based, piecewise-linear velocity fields that are not globally C^1?

---

> ### Author Response · Authors · 2025-11-21
> **Response to Reviewer vNfw**
>
> We thank the reviewer for their extremely positive assessment and for recognizing FALCON as a "mathematically elegant" and "practically useful" solution. We address the requests for clarification and discussion below:
>
> 1. **Practical guidance to mitigate numerical instability**: The reviewer is entirely correct that non-invertibility can have a significant impact on the quality of the likelihoods of the trained model. We took multiple steps to correct for, and address this potential issue through regularization to encourage invertibility:
> - **Regularization (Algorithm 1)**: We include a cycle-consistency term (reconstruction loss) that explicitly penalizes non-invertible trajectories. This implicitly mitigates conditioning issues of the Jacobian. This approach has been observed in the literature to improve conditioning in similar settings [1,2]. As shown in Fig. 6, there is a controllable trade-off: increasing this regularization improves the "invertibility" (and thus ESS/likelihood accuracy) at the cost of slight sample quality degradation. This provides a practical ``knob’’ for users to ensure sufficient numerical stability.
>  - **Empirical Invertibility (Appendix F.2)**: To quantify stability in “near-singular” regions, we trained a reverse FALCON model to invert the forward mapping. We observed invertibility reconstruction errors in the range of $10^{-4}$. This is on par with the numerical precision of strictly invertible discrete NFs (which also suffer from numerical error in practice), confirming that FALCON avoids singularity issues in relevant regions of the probability space.
>
> 2. **Smoothness requirements on $u_\theta$**: This is an interesting question. Investigating further, we find that $u_\theta \in C^1$ is sufficient as the reviewer suggests. However, $u_\theta$ which is $C^1$ *almost everywhere* is also sufficient. This includes ReLU-based, piecewise-linear maps that are not globally $C^1$. In this case, there will be a set of measure zero where the change of variables formula will not hold. In order to account for this case, we have generalized our theoretical result for the almost everywhere setting, which includes ReLU-based networks. This is acceptable for us as we will never hit this measure zero set in probability. We have added a remark to the theoretical section clarifying that while smooth activations (e.g., SiLU) might be preferred for optimization dynamics, the theoretical guarantees hold for piecewise-linear architectures as they satisfy this smoothness condition.
>
> ### References
> [1] Draxler, Felix, et al. "Free-form flows: Make any architecture a normalizing flow." International Conference on Artificial Intelligence and Statistics. PMLR, 2024.
>
> [2] Rehman, Danyal, et al. "Efficient Regression-Based Training of Normalizing Flows for Boltzmann Generators" arXiv preprint arXiv:2506.01158 (2025).

---

> > ### Comment · Reviewer_vNfw · 2025-11-27
> >
> > I would like to thank the authors for their detailed responses and careful analysis. After reading all reviews and the author rebuttal, I have decided to maintain my original score.
> >
> > This work removes an important obstacle for applying flow models to real scientific problems, and I believe it has clear methodological and practical significance. I also note that another reviewer pointed out the existence of a related work (arXiv:2506.01158). While I cannot completely rule out the possibility that the present submission may have been influenced by that preprint, I am more inclined to believe that the two studies arrived at similar ideas independently. The preprint focuses primarily on distilling a fast surrogate model from a pretrained flow model, which differs substantially in research motivation and scope from the current work. In addition, considering that the preprint has not yet been formally published, I evaluate the submitted manuscript as an independent contribution.
> >
> > Of course, if definite evidence were to show that the original manuscript borrowed from prior work without proper citation, then that would be a different matter. At present, however, I judge the contribution on its own merit.

---

### Official Review · Reviewer_cmBL · 2025-11-01

**Soundness:** 4
**Presentation:** 4
**Contribution:** 4
**Rating:** 8
**Confidence:** 4

**Summary:**

This paper describes FALCON (Few-step Accurate Likelihoods for Continuous Flows), which does what the title says. The addition of an invertability loss supports the training of a few-step flow map from which accurate likelihoods can be computed efficiently.  The results demonstrate this advances the state-of-the-art for Boltzmann Generators.

**Strengths:**

The results (in particular Figure 2) are impressive and advance the field.  Inference time is orders of magnitude faster than traditional CNFs.

Scalable Boltzmann generators can enable new direction in molecular modeling - this is an important advance.

The paper is well written and the approach is well motivated, both intuitively and mathematically.

**Weaknesses:**

Figure 4 is hard to interpret. What are the x's? Does FALCON really have a constant W2 regardless of the number of samples generated? I don't think this figure is clearly illustrating the data as intended.

Although the ability to use a larger model is attributed as a strength of the approach, for comparison purposes it would be nice to train a model of similar size to comparative approaches to factor the effect of model size on performance.

L_1 is used without introduction (defined in appendix, but not main paper).

**Questions:**

What do you mean by combining L and L_{avg}? This is unclear.

How could your approach be adapted to a transferable model (not single molecule system)?

---

> ### Author Response · Authors · 2025-11-21
> **Response to Reviewer cmBL**
>
> We thank the reviewer for their encouraging assessment and for recognizing the potential of FALCON to enable new directions in molecular modelling. We are particularly grateful for the suggestion to compare model sizes, which led to significant improvements in our results. We address the specific questions and clarifications below:
>
> 1. **Fig. 4 interpretation and formatting**: We thank the reviewer for pointing out the confusion in Fig. 4. The x-markers in the original figure were included purely for aesthetics and did not represent actual data points; they have been removed from the revised plot. The reviewer also correctly notes that FALCON should not exhibit a constant $\mathcal{E}$-$\mathcal{W}_2$ as the number of generated samples increases. In the original figure, the flat line for FALCON indicated the $\mathcal{E}$-$\mathcal{W}_2$ score obtained using 1e4 samples, whereas SBG’s curve reflected performance with an increasing number of samples. Although SBG improves with more samples, it never matches FALCON’s accuracy—even when drawing 250$\times$ more samples. Since FALCON’s $\mathcal{E}$-$\mathcal{W}_2$ performance also improves with more samples, the original visualization undercommunicated this. To address this, we have remade the figure, incorporating the following key changes:
> We now plot FALCON with an increasing number of samples, showing the expected decrease in $\mathcal{E}$-$\mathcal{W}_2$.
> We demonstrate that even if we sample 5e6 points with SBG, **the performance on $\mathcal{E}$-$\mathcal{W}_2$ is still not comparable to that of FALCON with 1e4 samples.**
>
> 2. **Efficiency of smaller architectures (model size comparison)**: We are particularly grateful for the suggestion to investigate smaller architectures. We trained a 3.2M parameter FALCON down from ~60M. As shown in the new Tab. 2 and Tab. 3 (summarized and consolidated for ease of access below), this lightweight model:
> - Scales effectively across all peptides tested.
> - Outperforms SBG (state-of-the-art discrete NF) on all metrics while enabling faster inference for larger peptides.
> Matches the performance of our originally reported larger models, showing that FALCON’s performance is driven by the method, and not parameter count.
> - These findings further underscore the limitations of ECNF++, which sees its performance degrade sharply with an increase in system size, highlighting that FALCON’s advantages stem from architectural enhancements and training loss rather than model parameter count.
>
> | Peptide | Model    | ESS ↑         | E–W₂ ↓         | T–W₂ ↓         |
> |---------|----------|---------------|----------------|----------------|
> | ALDP    | ECNF++   | 0.275 ± 0.010 | 0.914 ± 0.122  | 0.189 ± 0.019  |
> |         | SBG SMC  | —             | 0.741 ± 0.189  | 0.431 ± 0.141  |
> |         | FALCON   | 0.067 ± 0.013 | 0.225 ± 0.104  | 0.402 ± 0.021  |
> | AL3     | ECNF++   | 0.003 ± 0.002 | 2.206 ± 0.813  | 0.962 ± 0.253  |
> |         | SBG SMC  | —             | 0.598 ± 0.084  | 0.503 ± 0.029  |
> |         | FALCON   | 0.077 ± 0.004 | 0.544 ± 0.013  | 0.452 ± 0.011  |
> | AL4     | ECNF++   | 0.016 ± 0.001 | 5.638 ± 0.483  | 1.002 ± 0.061  |
> |         | SBG SMC  | —             | 1.007 ± 0.382  | 1.039 ± 0.069  |
> |         | FALCON   | 0.055 ± 0.003 | 0.686 ± 0.047  | 0.858 ± 0.077  |
> | AL6     | ECNF++   | 0.006 ± 0.001 | 10.668 ± 0.285 | 1.902 ± 0.055  |
> |         | SBG SMC  | —             | 1.189 ± 0.357  | 1.444 ± 0.140  |
> |         | FALCON   | 0.060 ± 0.017 | 0.892 ± 0.311  | 1.256 ± 0.132  |
>
>
> 3. **Definition of $L_1$** We appreciate the reviewer’s attentiveness to detail and have made minor modifications to the manuscript to ensure that all the appropriate terms are defined a priori. This term should read $L_{avg}$.
>
> 4.**Combining Losses** We have revised the text to clarify the relationship between the losses. This should have read combining $L_{cfm}$ and $L_{avg}$. We clarify that by adjusting the distribution of $s$ and $t$ in Equation (10), the general objective recovers $L_{cfm}$. This formulation allows us to combine $L_{cfm}$ and $L_{avg}$ in a unified framework.
>
> 5. **Transferability** This is an excellent question and the primary focus of future work. FALCON has clearly demonstrated high performance across global and local metrics on single-system peptides; ​the FALCON architecture (based on the DiT) is naturally suited for transferability—this can potentially be achieved through the introduction of conditioning on peptide and protein sequence, allowing for transferable training. We look forward to the potential of FALCON to scale to this setting.
>
> We reiterate our thanks for the high-quality review. The suggestion to investigate smaller models has substantially strengthened our manuscript's claims regarding efficiency.

---

> > ### Comment · Reviewer_cmBL · 2025-11-24
> >
> > The authors' response increases my already **high enthusiasm** for this paper.

---

### Official Review · Reviewer_ExRz · 2025-11-04

**Soundness:** 3
**Presentation:** 3
**Contribution:** 1
**Rating:** 4
**Confidence:** 4

**Summary:**

In this work, the authors tackle the problem of accurate likelihood estimation for the family of Flow Map models, an extension of standard CNFs. While efficient in terms of the number of steps needed for accurate sampling, Flow Maps cannot model the required change of variables needed for accurate density estimation.

The authors tackle this problem by introducing an additional regularization term that enforces the flow map to be approximately invertible (Eq. 8). If strong enough, this regularization term makes it possible to calculate the change in density using the log-determinant of the flow map’s Jacobian, similar to what is used in time-discrete normalizing flows. The authors formally show that the exactly invertible architecture is the minimizer of the proposed loss.

In their experimental evaluation, the authors show how a Flow Map trained using their proposed regularization technique can be used within the context of equilibrium sampling of peptide systems using importance sampling, similar to how standard CNFs and discrete-time NFs are used in Boltzmann generators. The experimental section focuses on showing improved scalability over CNFs and improved sample quality over discrete-time NFs. Furthermore, the paper studies the efficiency of the method (comparing training against discrete-time NFs and inference against CNFs) and studies the inference–accuracy trade-off. Throughout all experiments, the approach performs better than or on par with current state-of-the-art methods.

**Strengths:**

Overall, the paper is well executed. The background and preliminaries sections are well written and contain sufficient depth; the problem setting is clear, and the proposed solution is well described. Furthermore, the proposed solution itself seems intuitive (although novelty might be limited, as noted below). Finally, the experimental section is extensive and provides clear evidence in favor of the proposed approach.

As such, I am generally in favor of accepting the paper for publication based on the content alone. However, as highlighted below, I have a few concerns regarding the novelty of the paper that need to be answered before I can vote for acceptance with absolute confidence.

**Weaknesses:**

My primary objection to voting for acceptance of the paper at this time is the similarity of the proposed method to the one presented in Rehmann et al., 2025. This paper has a very similar objective: combining the benefits of CNFs and discrete-time NFs by enforcing approximate invertibility. Additionally, both papers share a very similar set of experiments (Rehmann et al. is, however, slightly more extensive). While the method presented here has, in theory, sufficient novelty to warrant publication, further clarification about the similarities and a comparison across methods are needed.

Without this clarification included in the paper, I do not feel confident in accepting the paper for publication.

In addition, I have included a short list of comments/questions below that I would appreciate the authors’ response to or see addressed in the paper:
- Fig. 1: This is very nit-picky, but the debiased target should completely overlap with the true density.
- Fig. 2: This should also include a discrete-time NF.
- Tab. 3: A key contribution of the paper is that, with the Flow Map–based approach instead of discrete-time NFs, FALCON should scale better. Table 3, however, shows that when considering large systems (e.g., Hexa-Alanine), the difference between the methods shrinks quite significantly, with SBG-IS and SBG-SMC even outperforming FALCON in terms of Wasserstein distance over energy distributions. This should be discussed in the paper.
- Tab. 4: It’s difficult to establish a clear trend in these results. ECNF is worse for smaller systems but has better training + inference time for Alanine Tetrapeptide. Conversely, the largest system is again considerably slower. SBG, on the other hand, is slower than FALCON for all system sizes except the largest. This should be discussed.

**Questions:**

See weaknesses

---

> ### Author Response · Authors · 2025-11-21
> **Response to Reviewer ExRz**
>
> We thank the reviewer for their thorough and insightful review. We are encouraged by the positive assessment of the paper's execution, soundness, and experimental breadth. We address the concerns regarding novelty and specific experimental trends below, with particular emphasis on the distinction from Rehman et al. (2025) and new results using optimized architectures.
>
> 1. **Comparison to Rehman et al. 2025 (RegFlow)**:  We respectfully clarify that while the two works both aim to improve the efficiency of Boltzmann Generators, they differ significantly in performance, architecture, approach and requirements, which we clarify in updated sections in the manuscript (Section 4.2, Section 5, Appendix C.2). We summarize the main clarifications here:
> - **Performance**: To evaluate relative performance, we added the best-performing RegFlow model (NSF) to Tab. 2 and 3. FALCON outperforms RegFlow on all metrics, demonstrating the empirical superiority of our approach.
> - **Architecture**: Conceptually, FALCON is the first few-step Boltzmann Generator to utilize a **free-form architecture** (DiT) to surpass state-of-the-art inherently invertible discrete flows. In contrast, RegFlow is constrained by the requirement of a strictly invertible architecture. By relaxing this constraint, FALCON achieves substantial performance gains through increased expressivity.
> - **Requirements**: RegFlow requires the specification of an invertible coupling (satisfied via pre-trained CNFs or static OT couplings). FALCON does not require this, allowing end-to-end training from scratch.
> Finally, in principle, it is possible to combine the two ideas by leveraging FALCON to define an invertible coupling to further distill into an invertible architecture, if stronger guarantees on invertibility are required.
>
> 2. **Fig. 1 overlap**: We appreciate the attention to detail. We clarify that while resampling converges to the target density in the infinite-sample limit, perfect alignment does not occur with finite samples. To ensure transparency regarding this distinction, we maintained the visual separation, but have updated Fig. 1’s caption to explicitly clarify that the lack of total overlap is due to the finite sample regime.
>
> 3. **Adding Fig. 2 discrete NF comparison**: As per the reviewer’s suggestion, we have revised Fig. 2 to include the best discrete NF (SBG). While discrete NFs allow for fast generation, they fail to match FALCON’s performance across all metrics. As shown in Fig. 4, even with 250$\times$ more samples, the state-of-the-art discrete NF cannot match FALCON’s performance, substantiating this claim.
>
> 4. **Tab. 3 FALCON scalability**: The reviewer correctly noted that the performance gap narrowed for larger systems in our initial submission, although FALCON still outperformed baselines overall. Additional analysis revealed that FALCON was substantially overparameterized for these tasks, which hindered convergence in these settings. Following Reviewer cmBL’s suggestion, we evaluated smaller architectures (3.2M parameters) for the rebuttal. We also use an improved regularization for larger systems, following the trends observed in Fig. 6 for larger systems (1e3 vs. 1e2). We find that with these optimized models, FALCON significantly outperforms SBG and ECNF++ on all global and local metrics for Hexa-Alanine (updated in Tab. 3). Crucially, FALCON (3.2M) outperforms SBG (64M) while using $\approx$ 95% fewer parameters, demonstrating that FALCON scales robustly while maintaining superior parameter efficiency.
>
> 5. **Tab. 4 training/inference efficiency**: We have updated Tab. 4 with the results from the new, efficient 3.2M parameter models. The trends are now much clearer:
> - **vs. SBG** With the new, smaller models (3.2M), we can better see that as we scale towards larger peptides, FALCON’s cumulative training and inference time is substantially less than that compared with SBG—this is true despite SBG leveraging a discrete NF, which is one-step generator that allows for fast inference, but suffers from slow training due to the MLE objective and larger size needed for higher expressiveness (64M).
> - **vs. ECNF++**: The efficiency gap is significant. Specifically, FALCON is significantly faster to train and run inference, largely because of the efficiency of the softly-equivariant DiT over ECNF++ architecture. Although the DiT achieves comparable performance on the dipeptide, the performance drop of ECNF++ as it scales is clearly observable, while FALCON achieves state-of-the-art performance on both global and local metrics.
> This highlights a key advantage: FALCON achieves the high expressivity of CNFs with inference speeds approaching discrete flows, avoiding the computational bottleneck of ECNF++.
>
> Overall, we thank the reviewer for their constructive comments. We hope that our revisions address all of their comments and concerns, and kindly request that the reviewer consider a revised score in light of these clarifications.

---

### Author Response · Authors · 2025-12-04

We would like to express our sincere appreciation to the reviewers for the care and time they invested in evaluating our submission. We are encouraged that the highly positive initial assessments highlighted the clarity of our exposition [ExRz, cmBL], the soundness of the methodology [ExRz, cmBL], the strength and breadth of our experimental evaluation [ExRz, cmBL, vNfw], and the potential high impact of the work [vNfw]. We first summarize the outcomes of the initial reviewer discussions then summarize how we addressed the major concerns raised by reviewers.
- Reviewer ExRz: Initial score 4. Indicated an openness to increase their score based on our response. However, no response was received before the discussion freeze.
- Reviewer cmBL: Initial score 8. Expressed “increased high enthusiasm” for the paper during discussion.
- Reviewer vNfw: Initial score 10. Maintained confidence in high score based on other reviewer comments.
- Reviewer tZEt: Initial score 6. Indicated a desire to increase score to 7, but as it did not exist, maintained the 6, but stated that this should be considered in the final decision.

During the rebuttal, we rigorously addressed all reviewer concerns, and are grateful that our additional clarification and analysis was well-received. We also extend our sincere thanks to both the previous and current ACs for their thoughtful engagement with the reviews and our responses. We summarize some of the main contributions and modifications made during the rebuttal phase below:

A key theme raised by reviewer cmBL concerned the efficiency and performance of smaller architectures. Acting on this suggestion, we conducted an extensive investigation of compact variants of our model, ultimately uncovering a notable result: a 3.2M-parameter version of FALCON—dramatically smaller than our original configuration and roughly comparable in size to SBG (60–90M)—performs exceptionally well. **This lightweight model often matched or exceeded the performance of the larger models originally included in our submission and also significantly surpassed larger discrete NF baselines**. Despite requiring more integration steps (2–16), this compact FALCON achieves wall-clock inference times comparable to the 1-step SBG baseline on the largest peptide (hexa-alanine) while outperforming it across all local and global metrics. These results also further revealed a key limitation of ECNF++ (an equivalently sized CNF), which performs strongly on alanine dipeptide but fails to scale effectively to larger systems—whereas an equivalently parameterized DiT model retains both scalability and accuracy.

In response to reviewer ExRz’s request for a clearer distinction between our approach and Rehman et al. (2025), we revised Table 1 and Appendix C.2 to explicitly articulate these differences. Most crucially, we added the strongest-performing RegFlow architecture (NSF) to Tables 2 and 3, revealing that **FALCON consistently outperforms RegFlow across all reported metrics**. Architecturally, FALCON is the first **few-step Boltzmann Generator to use a free-form architecture**, eliminating the structural constraints associated with strictly invertible discrete NFs. RegFlow, by contrast, requires inherently invertible architectures and the specification of an invertible coupling—either via pre-trained CNFs or static OT maps, whereas FALCON **enables fully end-to-end training without these dependencies**. These conceptual and empirical distinctions, now clearly delineated, underscore the novelty and advantages of our method.

All of these improvements and additions are reflected in the updated manuscript, particularly in Tables 2, 3, and 4. We believe the revised paper presents a clear, rigorous, and compelling contribution, and we remain highly confident in the strength and significance of our work.

---

### Meta-Review · Area_Chair_xQqt · 2025-12-30

**Summary:**

The paper investigates the application of continuous-time normalising flows (CNFs) as Boltzmann generators, where a key requirement for this task is the evaluation of the CNF's density. The paper first pointed out that acceleration methods like few-step flow models (e.g., mean flow), while improving on fast generation for CNFs, fails for likelihood evaluations due to the lack of invertibility guarantee. To address the issue, the paper introduces a regularisation loss to encourage invertibility of the learned few-step flow models. The paper also proposes mean flow's training objective in the context of Boltzmann generators (i.e., sampling from an unnormalised target density), as well as scalable implementations for it. Experiments on sampling benchmarks and comparisons with previous baselines suggest the efficacy of the proposed method.

While reviewers were generally happy for acceptance, one major concern was raised regarding the similarity of the approach to RegFlow. I believe this has been addressed by the author rebuttal. There are also other concerns regarding the proposed method's performance on larger systems, as well as some inconsistencies regarding the gains of the method across different experiments.

**Reviewer Concerns:**

Addressed concerns:
- Comparison with RegFlow regarding novelty. In reply, the authors stated the biggest difference between RegFlow and FALCON being the requirement of using an invertible neural network architecture. The AC agrees with this justification.

Concerns that are still outstanding:
- Invertibility is approximated achieved, no direct analysis regarding the impact of this degree of approximation.

**Reviewer Scores:**

Reviewer tZEt indicated a desire to increase score to 7 (which does not exist), so they are indicating somewhere between 6 (their initial score) and 8 (which is the next available score level).

Reviewer vNfw has already maxed out their score (10).

Reviewer ExRz's major concern, i.e., similarity to RegFlow, would have been viewed as addressed. They have other concerns regarding experiments, for which I don't know whether the author rebuttal is sufficient to address. Therefore, I cannot predict the potential score change though.

---

### Decision · Program_Chairs · 2026-01-26

Accept (Oral)